# Dissolved organic nutrients dominate melting surface ice of the Dark Zone (Greenland Ice Sheet)

**Alexandra T. Holland[1], Christopher J. Williamson[1,2], Fotis Sgouridis[4], Andrew J.**
**Tedstone[1], Jenine McCutcheon[5], Joseph M. Cook[6], Ewa Poniecka[7], Marian L. Yallop[2],**
**Martyn Tranter[1], Alexandre M. Anesio[1,3], and The Black & Bloom Group\***
[1]Bristol Glaciology Centre, School of Geographical Sciences, University of Bristol, Bristol, BS8 1HB, UK.
[2]School of Biological Sciences, University of Bristol, 24 Tyndall Avenue, Bristol, BS8 1TQ, UK.
[3]Department of Environmental Science, Aarhus University, Roskilde, 4000, Denmark.
[4]School of Geographical Sciences, University of Bristol, Bristol, BS8 1RL, UK.
[5]School of Earth and Environment, University of Leeds, Leeds, LS2 9JT, UK.
[6] Department of Geography, University of Sheffield, Winter Street, Sheffield, S3 7ND, UK
[7]School of Earth and Ocean Sciences, Cardiff University, Main Building, Park Place, Cardiff, CF10 3AT, UK.
*A full list of authors and their affiliations appears at the end of this paper.
*Correspondence to:* Alexandra T. Holland (Alexandra.holland@brisol.ac.uk)
**Abstract.** Glaciers and ice sheets host abundant and dynamic communities of microorganisms on the ice surface
(supraglacial environments). Recently, it has been shown that Streptophyte glacier algae blooming on the surface ice
of the south-west coast of the Greenland Ice Sheet are a significant contributor to the 15-year marked decrease in
albedo. Currently, little is known about the constraints, such as nutrient availability, on this large-scale algal bloom.
In this study, we investigate the relative abundances of dissolved inorganic and dissolved organic macronutrients (N
and P) in these darkening surface ice environments. Three distinct ice surfaces, with low, medium and high visible
impurity loadings, supraglacial stream water and cryoconite hole water were sampled. Our results show a clear
dominance of the organic phase in all ice surface samples containing low, medium and high visible impurity
loadings, with 93% of the total dissolved nitrogen and 67% of the total dissolved phosphorus in the organic phase.
Mean concentrations in low, medium and high visible impurity surface ice environments are 0.91 µM, 0.62 µM and
1.0 µM for dissolved inorganic nitrogen (DIN), 5.1 µM, 11 µM and 14 µM for dissolved organic nitrogen (DON),
0.03 µM, 0.07 µM and 0.05 µM for dissolved inorganic phosphorus (DIP) and 0.10 µM, 0.15 µM and 0.12 µM
dissolved organic phosphorus (DOP) respectively. DON concentrations in all three surface ice samples are
significantly higher than DON concentrations in supraglacial streams and cryoconite hole water (0 µM and 0.7 µM,
respectively). DOP concentrations are higher in all three surface ice samples compared to supraglacial streams and
cryoconite hole water (0.07 µM for both). Dissolved organic carbon (DOC) concentrations increase with the amount
of visible impurities present (low: 83 µM, medium: 173 µM and high: 242 µM) and are elevated compared to
supraglacial streams and cryoconite hole water (30 µM and 50 µM, respectively). We speculate that the architecture
of the weathering crust, which impacts on water flow paths and storage in the melting surface ice, and/or the
production of extracellular polymeric substances (EPS), containing both N and P in conjunction with C, is
responsible for the temporary retention of DON and DOP in the melting surface ice. The usual presence of
measurable DIP and DIN, principally as $NH_4^+$, in the melting surface ice environments, suggests that factors other
than macronutrient limitation are controlling the extent and magnitude of the glacier algae.

## 1. Introduction

There has been a significant increase in the net mass loss of the Greenland Ice Sheet (GrIS) during the past two
decades (Rignot and Kanagaratnam, 2006;Rignot et al., 2011;Shepherd et al., 2012), from 34 Gt yr[-1] to 215 Gt yr[-1]
between 1992 and 2011 respectively (Sasgen et al., 2012). Surface melt is  the primary driver for the increase in ice
mass loss (~68%) since 2009, with the remaining (~32%) coming from solid ice discharge or calving (Enderlin et
al., 2014). There are two major reasons for this marked increase in surface melting. First, the extent of bare, melting
surface ice increased, on average, by 7158 km[2] per year from 2000 to 2014 (Enderlin et al., 2014;Shimada et al.,
2016).  Second, the albedo of bare surface ice areas declined between 2000 and 2012, with south-west Greenland
exhibiting the greatest decrease of up to 18% (Box et al., 2012). A persistent Dark Zone in this region, some 20-30
km inland and ~50 km wide, has reoccurred annually since at least 2001 (Wientjes and Oerlemans, 2010;Box et al.,
2012;Stroeve et al., 2013;Tedstone et al., 2017). There is significant variability in the annual extent of the Dark Zone
(Shimada et al., 2016), which may be the result of both inter-annual climatic variability and factors associated with
the ice surface, such as melt-out of ancient Holocene dust particles (Wientjes et al., 2011;Tedstone et al., 2017).
Both snow and bare ice albedo are reduced by light absorbing impurities (LAIs), of both biological and
mineralogical origin (Gardner and Sharp, 2010), which include atmospheric dust and black carbon, cryoconite, and
particulates within the meteoric ice that melt out during the ablation season (Warren and Wiscombe, 1980;Warren,
1984;Warren and Wiscombe, 1985;Gardner and Sharp, 2010;Wientjes et al., 2012;Cook et al., 2016a). The
importance of biological LAI, particularly Streptophyte glacier algae, which bloom in surface ice environments
during summer ablation seasons, as a factor in albedo decline has been identified in recent years (Yallop et al.,
2012). The  effect has become known as "bioalbedo", which is derived from the original term "biological albedo
reduction" (Kohshima et al., 1993;Cook et al., 2017a). Bioalbedo is attributed to a combination of the high
abundance of cells that grow during the bloom (up to ~10[4] cells ml[-1] surface ice) and the heavily pigmented nature
of ice algal cells, which include a unique dark UV-VIS absorbing pigment, purpurogallin, that provides photo-
protection from the extreme solar radiation in supraglacial environments (Remias et al., 2012;Williamson et al.,
2018). Tedstone et al., (2017) concluded that ice algal blooms are the main factor responsible for inter-annual
variability in the extent, magnitude and duration of the Dark Zone, which seem to be regulated by climatic drivers,
including the June-July-August sensible heat flux anomaly and the timing of snow-line retreat. The spatial extent of
ice algal blooms may also be linked to the availability of mineralogic LAIs, such as late Holocene dust particles
melting out of the meteoric ice (Wientjes et al., 2012). However, the linkage between particles and algae is not
presently understood (Tedstone et al., 2017).
C, N and P are essential for all living organisms, providing the basis for cellular mass and all metabolic activity
(Redfield et al., 1963;Hessen et al., 2013). Carbon is usually in ready supply in surface ice environments, both from
the atmosphere and from bubbles trapped in snow and ice, and so nitrogen and phosphorus are more likely the
limiting factors for growth and activity of microorganisms (Stibal et al., 2009;Lutz et al., 2017). Bioavailable forms
of N are less readily available, being largely confined to $NO_3^-$ and $NH_4^+$ in dry and wet deposition from the
atmosphere (Wolff, 2013), and from snow- and ice-melt (Telling et al., 2011). Dissolved inorganic phosphorous
(DIP) is typically the least available nutrient in supraglacial environments, since it is a largely rock-derived and is
only released by chemical weathering or bio-mining (Stibal et al., 2008b;Stibal et al., 2009) . P sources in remote
glaciated environments, such as the Dark Zone, are largely confined to the small quantities of particulates deposited
from the atmosphere and the melt out of debris in snow and ice (Wientjes and Oerlemans, 2010).
The presence of such large-scale algal blooms in the Dark Zone might suggest that these environments are nutrient-
rich. This would contrast with the current literature, which suggests that supraglacial environments in the Dark
Zone, similar to those found in Svalbard, the margins of the Greenland Ice Sheet and Antarctica, are extremely
oligotrophic (Stibal et al., 2008b;Stibal et al., 2009;Telling et al., 2011;Telling et al., 2012;Bagshaw et al.,
2013;Hawkings et al., 2016;Wadham et al., 2016). Mean dissolved inorganic nitrogen (DIN) concentrations in
Greenland ice are ~1.4 μM, with $NO_3^-$ and $NH_4^+$ composing 0.97 μM and 0.39 μM, respectively (Wolff , 2013).
There are relatively few measurements of nutrient concentrations in the surface ice environments of the Dark Zone
(Telling et al., 2012;Wadham et al., 2016), but the average $NO_3^-$ concentration in surface ice along the K Transect
east of Kangerlussuaq, which passes through the Dark Zone, has been reported to be $0.6 \pm 0.1$ μM between 17-79
km from the ice sheet margin (Telling et al., 2012), while DIP concentrations were below the detection limit,
0.33μM P (Telling et al., 2012). DIN concentrations in snow sampled before the start of the ablation season at the
margin of the GrIS had higher concentrations, with an average of 1.4 μM (Telling et al., 2012), similar to those of
Wolff (2013). Hence, there is no real evidence that neither N nor P concentrations in snow and ice sampled in the
vicinity of the Dark Zone are higher than for average Greenland ice.  The relatively low concentrations of
macronutrient in the snow and ice of the SW Greenland Ice Sheet means that algal blooms are likely to rapidly
sequester N and P from snow and ice melt, particularly as the blooms reach their zenith at the height of the ablation
season. For example, NPP (Net Primary Production) values in the wet, melting surface ice (also called rotten ice, or
the weathering crust) during blooms range from $21 - 100$ μmol C $l^{-1}$ $day^{-1}$ (Chandler et al., 2015;Williamson et al.,
2018). Should the mean DIN concentration of the ice melt be 1.4 μM, this implies a C:N molar ratio of $15 - 71$ if all
the DIN is sequestered into new organic matter and no other sources of DIN are present. There is no readily
available C:N ratio of glacier algae in the literature, but typical C:N ratios of sea ice algae are in the range of 12-46
(Niemi and Michel, 2015). It is even more difficult to find C:N:P ratios of glacier algae, but should the C:P ratio be
in the region of 100:1 to 1000:1, the P demand will be $0.02 - 1$ µM.
Blooms in other aquatic ecosystems are associated with efficient recycling of nutrients when new sources of N and P
are in scarce supply, often with a balance between nutrient uptake and remineralization (Dodds, 1993), allowing
nutrient accumulation in biomass over time. This balance does not appear to arise in the surface ice environments of
other High Arctic and polar glaciers studied to date. These are predominantly in cryoconite holes, which are water-
filled cylindrical holes with an organic-rich basal sediment in the ice surface that host a range of microbes, including
cyanobacteria (Christner et al., 2003;Anesio and Laybourn-Parry, 2012;Telling et al., 2012). Dissolved
macronutrients tend to become concentrated in organic phases (Stibal et al., 2008b;Telling et al., 2014), suggesting
an imbalance in the uptake and remineralization of dissolved inorganic nutrients in cryoconite hole environments.
Indeed, the only ratio of primary production to remineralization measured in the Dark Zone is 30:1 (Nicholes et al.,
2019). To date, dissolved organic nitrogen (DON) concentrations in the Dark Zone have only been reported in two
studies (Telling et al., 2012;Wadham et al., 2016), but neither focus on ice populated by Streptophyte glacier algae.
Telling et al., (2012) reported a near 1:1 relationship between $NO_3^-$ and total dissolved nitrogen (TDN), suggesting
that DON comprised only a small portion of the TDN pool in snow and ice samples. By contrast, Wadham et al.,
(2016) suggested mineralization of organic matter by microbial activity, either within the cryoconite holes
themselves or in debris- and cryoconite-rich "dirty" surface ice contributed to DON concentrations that could reach
0.7 µM and 3.0 µM, respectively. No dissolved organic phosphorous (DOP) concentrations in the surface ice
environments in the Dark Zone have been reported to date.
Several studies have noted the heterogeneity in the spatial distribution of glacier algae in the melting surface ice of
the Dark Zone (Yallop et al., 2012;Williamson et al., 2018). This heterogeneity occurs on length scales of cm to 10s
of m (Yallop et al., 2012). This might well signify that macronutrient concentrations are also variable on this scale,
yet no studies to date have examined variability on these length scales. We contend that it is important to determine
the concentrations and relative proportions of dissolved inorganic and organic nutrients in melting surface ice
environments of Dark Zone, particularly during Streptophyte glacier algae blooms, since a knowledge of both DIN,
DON, DIP and DOP may be crucial to better understand how glacier algae and bacteria can retain, utilize and
recycle their limited nutrients to sustain the large-scale blooms observed in this region of the Greenland Ice Sheet.
The aims and objectives of this study, therefore, are threefold. First, we aim to quantify dissolved nutrient
concentrations in the supraglacial environments of the Dark Zone during the peak ablation season. Second, we
determine the relative abundance of dissolved inorganic and organic nutrients during the peak ablation season when
microbial recycling is likely to have the greatest influence on the dissolved inorganic and organic ratios. Finally, we
investigate if there are systematic changes in the relative proportions of dissolved macronutrients during increased
colonization of melting surface ice, which might shed light on the limiting nutrient on algal blooms.

**2. Methods**

## 2.1 Field Site and Sampling

A field camp was established within the Dark Zone, adjacent to Kangerlussuaq, during the summer of 2016. The camp was located approximately 30 km inland from the ice margin, near to the 'S6' weather station on the K-transect (Fig 1; 67°04'43.3" N, 49°20'29.7" W). Samples were collected from a designated area of approximately 500 × 500 m, which included surface ice, supraglacial stream and cryoconite hole habitats. Sampling occurred at intervals of approximately three days from 15th of July to 14th of August 2016. A categorical sampling strategy was employed, given the evident spatial heterogeneity apparent in ice algal distributions. Five different habitats were sampled; melting surface ice with three differing amounts of visible impurities, referred to here as surface ice with "low" ($n=19$), "medium" ($n=19$), and "high" ($n=19$) visible impurities) (Fig. 2) (Yallop et al., 2012). Water from supraglacial streams ($n=10$) and cryoconite holes ($n=14$) was randomly collected, both to act as a comparison with the melting surface ice and to examine how dissolved nutrients were transported through the weathering crust, which is the melting layer of surface ice that has a different physical architecture to the underlying ice (Fig. 2). Surface ice habitats were sampled from a 1 × 1 m area chosen at random, from which the top ~2 cm of ice was removed using a pre-cleaned ice saw.

Samples from all five categories were collected for the analysis of dissolved inorganic and organic nutrients and dissolved organic carbon (DOC). Algal cell abundances were determined on surface ice samples only. Ice collected for nutrient analysis and algal cell abundance was placed into a clean/sterile Whirl-pak™ bag, while that collected for DOC analysis was transferred into a glass jar that was first rinsed three times with sample. Ice samples were left to melt overnight in the lab tent, typically taking 4-5 h. Supraglacial stream water samples for nutrient analysis were collected using high-density polyethylene plastic bottles (Nalgene™), whereas those for DOC analysis were collected in glass jars. Both sampling containers were rinsed three times with sample prior to collection. Cryoconite hole water used for nutrient and DOC analysis was collected using a large pipette and transferred into a Nalgene™ bottle or glass jar, respectively. The large pipette and collection vessels were rinsed three times with sample prior to collection. All high-density polyethylene plastic bottles (Nalgene™) for nutrient samples were acid washed in ~10% HCl solution prior to first use and all glass jars for DOC samples were furnaced at 500°C for four hours prior to first use.

Some 15 ml of the homogenised, unfiltered ice melt and water samples were subsampled and fixed using 25% glutaraldehyde at 2% final concentration for quantifying algal cell abundance. These fixed samples were stored outside in the dark at ambient ice sheet temperatures. Ice melt and water samples for nutrient analysis were filtered through a 25 mm, 0.22 μm cellulose nitrate inline syringe filter (Whatman™) and stored in high density polyethylene plastic bottles (Nalgene™; 30mL). The bottles were immediately frozen and stored at a temperature of -20°C, using a Waeco 32L Freezer. Ice melt and water samples for DOC analysis were filtered using a glass filtration column and a furnaced 47 mm, 0.7 μm GF/F. The filtration column was washed three times with sample water prior to collection of the filtrate. The filtrate was stored in pre-furnaced amber glass vials and acidified with 100 μL of 1M HCl. They were chilled to a temperature of ~3°C by storing the samples in a box at ambient air temperature. The samples were maintained at this temperature during transport and in storage at the LowTex

Laboratory at the University of Bristol. Nutrient samples were thawed immediately prior to analysis using a ~40°C
hot water bath. Procedural blanks (*n=9*) were collected over the course of the sampling season, by processing
deionised water in place of sample.
**2.2 Analytical Methods**
Algal cell abundance was quantified using a Fuchs-Rosenthal haemocytometer (Lancing, UK) on a Leica DM 2000
epifluorescence microscope with attached MC120 HD microscope camera (Leica, Germany). For samples
containing sufficient cell abundance, a minimum of 300 cells were counted to ensure adequate assessment of
assemblage diversity (Williamson et al., 2018).
DIN species include $NH_4^+$, $NO_2^-$ and $NO_3^-$ and were quantified as follows. First, $NH_4^+$ was quantified
spectrophotometrically using a Lachat QuickChem® 8500 Series 2 Flow Injector Analyzer (FIA; QuickChem®
Method 31-107-06-1-I). Measurements were based on a phenolate-hypochlorite alkaline reaction method measured
at 630nm (Solorzano, 1969). The limit of detection (LoD) was 0.62 µM, determined by dividing the standard
deviation of the response of the calibration curve by the slope of the calibration curve, then multiplying the result by
3 (Shrivastava and Gupta, 2011). Precision was ±2.1%, and accuracy was +8.5%, as determined from comparison
with a gravimetrically diluted 1000 mg L$^{-1}$ $NH_4^+$-N certified stock standards to a concentration of 1.1 µM. (Sigma
TraceCERT®). Second, $NO_2^-$ and total oxidised nitrogen (TON) ($NO_2^-$ + $NO_3^-$) were quantified
spectrophotometrically using a Gallery Plus Automated Photometric Analyzer (Thermo Fisher Scientific, UK). This
combination of analysis allows the original $NO_3^-$ concentration to be determined by subtracting $NO_2^-$ from TON.
TDN is the sum of DIN and DON, and was determined by digesting the samples with a potassium persulfate,
sodium hydroxide and boric acid reagent and autoclaving at 121°C for 30 minutes (Grasshoff et al., 1999). This
process causes the oxidation of organic nitrogen compounds, which can then be measured as TON as above.
Purification of the potassium persulfate was conducted via recrystallisation in order to remove any N contamination.
Measurements were based on the hydrazine-sulfanilamide reaction method measured at 540nm. DON was then
estimated by subtracting DIN from TDN (i.e. DON= TDN-DIN). LoD were 0.14 µM ($NO_2^-$), 0.64 µM (TON) and
0.87 µM (TDN/DON). Precision was ±0.87% ($NO_2^-$), ±1.17% ($NO_3^-$) and ±0.63% (TDN/DON), and accuracy was -
4.04% ($NO_2^-$), -8.07% ($NO_3^-$) and -5.7% (TDN/DON), as determined from comparison with gravimetrically diluted
1000 mg L$^{-1}$ $NO_2$—N and $NO_3$—N certified stock standards to a concentration of 0.71 µM ($NO_2^-$), 1.4 µM ($NO_3^-$) and
7.1 µM (TDN/DON) (Sigma TraceCERT®).
TDP (total dissolved phosphorus) is the sum of DIP (principally $PO_4^{3-}$) and DOP. The same persulfate digestion
method described for TDN was used to measure TDP as $PO_4^{3-}$. $PO_4^{3-}$ in both undigested and the digested samples
was quantified using a Lachat QuickChem® 8500 Series 2 Flow Injector Analyzer (FIA; QuickChem® Method 31-
115-01-1-I) using the molybdenum blue method measured at 880nm. DOP was determined by the subtraction of DIP
in the undigested sample from the TDP in the digested sample (i.e. DOP = TDP-DIP). The LoD was 0.02 µM ($PO_4^{3-}$
and TDP/DOP). Precision was ±1.6% ($PO_4^{3-}$) and ±3.1% (TDP/DOP), and accuracy was +2.3% ($PO_4^{3-}$) and +5.0%
(TDP/DOP), as determined from comparison with gravimetrically diluted 1000 mg L$^{-1}$ PO$_4$-P certified stock
standards to a concentration of 0.65 μM (Sigma TraceCERT®).
DOC concentrations were quantified using a Shimadzu TOC-L Organic Carbon Analyzer, with a high sensitivity
catalyst. Non-purgeable organic carbon (NPOC) was measured after acidification of samples with HCl and catalytic
combustion (680°C) of dissolved organic carbon to carbon dioxide, which was then measured by infrared
absorption. The LoD was 9.5 μM. Precision was ±2.4% and accuracy was -5.9%, as determined from comparison
with gravimetrically diluted 1000 mg L$^{-1}$ TOC certified stock standards to a concentration of 83.3 μM (Sigma
TraceCERT®).
**2.3 Data Analysis**
All measurements below the LoD were considered to be 0 for all statistical analyses. All DIN, DON, DIP, DOP and
DOC data were water blank-corrected using values from the respective field procedural blanks (Table 1).
Additionally, all blank corrected values that were negative were assumed to be 0 for all statistical analyses.
Statistical analysis was performed in RStudio v.1.1.414 (RStudio, Inc 2018). Identification of statistical differences
between the nutrient content, DOC concentration and algal cell abundance in different habitats was achieved using
1-way analysis of variance (ANOVA) or t-test comparisons, with post-hoc Tukey HSD analysis applied to all
significant ANOVA results. Pearson's product-moment correlations were used to identify correlations between
DON, DOC and algal cell abundance. Homogeneity of variance and normality of distribution were tested prior to all
parametric analyses, and model assumptions were verified by examination of model criticism plots.

**3. Results**
**3.1 Dissolved nutrient concentrations in surface ice with differing levels of visible impurities**
Supraglacial environments are extremely oligotrophic, making measurements of dissolved nutrients difficult.
Dissolved nutrient concentrations reported in previous studies of supraglacial environments are typically below or
just above instrument limit of detections. Some 54 DON, 41 DIN, 74 DOP, 40 DIP and 59 DOC samples out of a
total of 81 samples for all five supraglacial habitats had concentrations above the LoD.
Dissolved organic concentrations were significantly higher than dissolved inorganic concentrations for nitrogen and
phosphorus. Some 93% of the TDN was in the form of DON and about 67% of TDP was present in the form of DOP
in all three surface ice habitats. Mean DON concentrations for the three surface ice habitats range from 5.1-14.0 μM,
while those for DIN range from 0.62-1.0 μM (Fig. 3, Table 1). Overall, mean DON concentrations for the three ice
surface habitats, were significantly higher ($F_{1,71}$=12.4, $p<0.0001$) than mean DIN concentrations. Similarly, DOP
concentrations were usually at least twice those of DIP concentrations for the three ice surface habitats, with mean
values ranging from 0.10-0.15 μM and 0.03-0.07 μM respectively (Fig. 4, Table 1). T-tests revealed significant
differences between DON and DIN in all three surface ice habitats (low: $t_{36}=3.6$, $p<0.001$, medium: $t_{36}=5.3$,
$p<0.0001$, high: $t_{36}=7.4$, $p<0.0001$, (Fig. 3) and DOP concentrations as significantly higher than DIP concentrations
for all three surface ice habitats (low: $t_{36}=3.1$, $p<0.01$, medium: $t_{36}=2.1$, $p<0.05$, high: $t_{36}=3.7$, $p<0.001$) (Fig. 4).
DON and DOC concentrations in the three surface ice habitats showed clear trends with increasing visible impurities
(Fig. 3 & 5). DON concentrations increased significantly from low to medium and low to high visible impurity
loadings ($F_{4,71}=19.8$, $p<0.05$, $F_{4,71}=19.8$, $p<0.001$, respectively), while DOC concentrations increased significantly in
ice with high and low visible impurity loading ($F_{4,74}=6.8$, $p<0.01$).

### 3.2 Links between algal abundance and dissolved organic nutrients

Algal cell abundance, which ranged from 90 cells ml$^{-1}$ to $0.98 \times 10^4$ cells ml$^{-1}$, increased significantly with the
amount of visible impurities seen on the ice surface, as shown in Figure 6 ($F_{2,54}=26.1$, $p<0.0001$). No determination
of the mineralogic composition of the visible impurities was conducted. A Pearson's product-moment correlation
was undertaken to illustrate the relationship between average algal abundance and average DOC and DON
concentrations, as DOC and DON concentrations also increased with the amount of visible impurities present.
Correlations between average algal cell counts versus both DON and DOC surface ice concentrations were
significant ($t_3=3.5$, $p<0.05$, $r=0.9$ and $t_3=5.4$, $p<0.01$, $r=0.95$, respectively). Comparison of DOP surface ice
concentrations and algal cell counts were not significant.
Dissolved organic nutrient ratios were assessed to investigate the presence of a limiting nutrient. Molar DON:DOP
ratios, ranging from 49 to 120, were elevated for all three surface ice environments compared to the 16:1 Redfield
Ratio, and DOC:DOP ratios for all three surface ice habitats, which ranged from 800 to 2000, were considerably
higher, as much as ~19 times the Redfield ratio, 106:1 (Table 1). DOC:DON ratios, which ranged from 16 to 17,
were, on average, twice the balanced 6.6:1 ratio (Table 1). DON:DOP and DOC:DOP ratios also increased with the
amount of visible impurities present, while DOC:DON ratios remain relatively constant for the three surface ice
habitats (Table 1).

### 3.3 Low transport of dissolved organic nutrients within the water table

Mean DON and DOP concentrations were significantly lower in supraglacial streams (0 µM and 0.07 µM,
respectively) and cryoconite hole water (0.7 µM and 0.07 µM, respectively) compared to low, medium and high
visible impurity ice. All DON concentrations for cryoconite hole and supraglacial stream water were below the LoD
(Fig. 3). DIN concentrations were relatively constant over all supraglacial habitats with mean concentrations ranging
from 0.62 µM to 1.0 µM. DOP concentrations in supraglacial stream ($0.07 \pm 0.03$ µM) and cryoconite hole water
($0.07 \pm 0.02$ µM µM) were not significantly different from DIP concentrations ($0.01 \pm 0.01$ µM and $0.06 \pm 0.02$ µM,
respectively). DIP concentrations in low ($0.03 \pm 0.02$ µM), medium ($0.07 \pm 0.02$ µM) and high ($0.05 \pm 0.01$ µM)
visible impurity ice were only slightly elevated compared to supraglacial streams, whereas cryoconite hole water
concentrations were comparable to the three surface ice habitats. Mean DOC concentrations in supraglacial stream
and cryoconite hole water (30 µM and 50 µM, respectively) were significantly lower than ice with high visible
impurities ($F_{4,74}=6.8$, $p<0.001$, in both cases) (Fig. 5).

## 4. Discussion

### 4.1 Dominance of dissolved organic over dissolved inorganic phases in melting ice surface environments

Dissolved organic nutrients (DON and DOP) dominate dissolved inorganic nutrients (DIN and DIP) in the melting surface ice environments of this region of the Dark Zone (Fig. 3 & 4), in contrast with the dominance of DIN in studies of fresh snow or ice cores from the GrIS (Telling et al., 2012;Wolff, 2013), which has a mean concentration of 1.4 $\mu$M. Further, DIN also dominates on the margins of the ice sheet, where Telling et al., (2012) found DIN concentrations in snow to be 1.4±0.2 $\mu$M, but DON concentrations to be non-detectable. Wadham et al., (2016) reported relatively similar DIN (1.3 $\mu$M) and DON (~1.6 $\mu$M, assuming DON = TDN-DIN in their tabulated data) concentrations in a small number (n = 7) surface, debris-rich ice in the Dark Zone of the GrIS during the main ablation season., but these values were thought likely to be associated with dispersed cryoconite, the dark organic-rich sediment that accumulates in the bottom of cryoconite holes and larger supraglacial water bodies. Otherwise, DON was not measurable in snow and surface ice, prior to melting. In summary, this suggests that potential input of dissolved N-species to supraglacial environments from fresh snow and melting meteoric ice are dominated by DIN, rather than DON. There is too little data on DIP and DOP to be confident that the this is also the case for P species. By contrast, dissolved N-species in the melting ice surface of the Dark Zone shifts to a domination of DON during the peak ablation season (July and August), when blooming of glacier algae occurs. We therefore hypothesise that the algae present in these blooms drive the shift in nutrients during the peak ablation season from the dissolved inorganic phase to the dissolved organic phase.

### 4.2 Association of dissolved organic nutrients and algal abundance

Figure 6 shows that algal abundance increases in the ice with low, medium and high visible impurities. The blooming of the algal cells is also associated with trapping of other mineral particulates at the surface. Yallop et al., (2012) reported a 3:2 mineral particle to algal cell ratio for surface ice collected in the Dark Zone, although these particles have only a minor impact on the albedo reduction at the surface (Cook et al., 2019). It is clear from Fig. 3 that the mean DON concentration increases from low to high visible impurities, consistent with DON formation being linked to glacier algae blooms. This is most likely due to a combination of extracellular exudation of polymeric substances and the decomposition of glacier algal cells within the supraglacial system. Concentrations of $NO_3^-$ and $NO_2^-$ are zero (Table 1), and $NH_4^+$ is the only measurable DIN species (mean values range from 0.6 to 1 $\mu$M). The absence of measurable $NO_3^-$ and $NO_2^-$ is consistent with the uptake of these species by glacier algae, and the emergence of $NH_4^+$ as the dominant DIN species is consistent with heterotrophic remineralization of organic matter (Telling et al., 2012). We note that the mass of N held in the microbial biomass is likely increasing over time, since the sum of the mean DIN and DON concentrations (1.0 $\mu$M and 0.0 $\mu$M respectively) in the supraglacial stream water, which is the ultimate sink of macronutrients from the melting ice surface, is less than the average DIN

concentration of the melting ice (1.4 μM) (Telling et al., 2012;Wolff, 2013;Wadham et al., 2016). The only
measurable DIN species in supraglacial meltwater is $NH_4^+$, which points to ammonification being an important
process in terms of N dynamics and loss of labile N from the melting surface ice. Previous studies of the relative
rates of primary production and bacterial production in both the margins and the Dark Zone have produced ratios of
30:1 (Yallop et al., 2012;Nicholes et al., 2019). The dominance of dissolved organic nutrients and $NH_4^+$ in surface
ice environments documented here, in combination with reduced secondary production relative to net primary
production in the same environments, indicates an inefficiency in the microbial loop for remineralization of organic
nutrient N-stocks (Fig. 7).
There is less data in the literature on the relative abundance of DIP and DOP in snow and ice, but here we show that
there are similarities between the accumulation of dissolved N into DON and P into DOP species (Fig. 3 & 4). Mean
DOP concentrations in the five sampled environments are higher than mean DIP. This is consistent with uptake of P
by glacier algae and subsequent recycling of P into organic forms. An interesting observation is that there is usually
some measurable DIP found in the particulate-rich environments that were sampled (surface ice and cryoconite hole
water), whereas the mean DIP in the supraglacial stream water is the lowest value recorded and below the limit of
detection. This suggests two things. First, that particulates are the source of DIP, and second, that export of P from
the melting surface ice is largely by DOP. We noted above that particulates are associated with glacier algae in the
melting surface ice, and it appears that as algal blooms develop, more particulates become trapped in the surface
layer (Yallop et al., 2012). It may well be that there is an "inorganic symbiosis" between the glacier algae and the
trapped particulates, which provide a P source for algal growth.
Our results on the dominance of DON and DOP are consistent with findings of previous studies in polar glacier
surface aquatic environments (Stibal et al., 2008a;Stibal et al., 2008b;Stibal et al., 2009;Wadham et al., 2016). For
example, Stibal et al., (2008) reported that DON (~72%) and DOP (~89%) in waters in cryoconite holes on a
Svalbard glacier dominated the total dissolved N and P pools. Wadham et al., (2016) found elevated DON
concentrations in water in cryoconite holes and debris-rich surface ice in the Dark Zone, suggesting either
mineralization of organic matter by microbial activity or leaching of allochthonous organic matter in debris. These
observations suggest that conversion of dissolved inorganic to dissolved organic nutrients by microbial communities
in melting surface ice environments may be a common process on glacier surfaces
**4.3 Retention of nutrients at ice sheet surface**
The low concentration of DIN, DIP, DON and DOP in the supraglacial meltwaters relative to the melting surface ice
suggests that macronutrients are retained in these surface environments. Melting ice surfaces in the Dark Zone often
have a veneer of low density, wet porous ice, which may reach depths of 1-2 m, known as the "weathering crust"
(LaChapelle, 1959;Müller and Keeler, 1969;Munro, 1990;Irvine-Fynn et al., 2012). The intense short wave radiation
during summer often causes internal melt along ice crystal boundaries, resulting in a surface ice layer with
heterogeneous thickness, density, porosity and water content (Müller and Keeler, 1969;Cook et al., 2016b;Christner
et al., 2018). The porous nature of the weathering crust allows flow paths to form through the water table that exists
within the surface ice (Irvine-Fynn et al., 2012;Cook et al., 2016b;Rassner et al., 2016;Christner et al., 2018), which
act as important links between different supraglacial environments and are believed to transport microbes and
nutrients via subsurface flow (Irvine-Fynn et al., 2012;Hoffman et al., 2014;Karlstrom et al., 2014;Cook et al.,
2016b). Water is often in temporary storage in the weathering crust (Irvine-Fynn et al., 2012), particularly at depth
where connectivity of flow paths can be low. It follows that the first explanation for retention of dissolved organic
nutrients in the weathering crust is that they accumulate in water stored in the weathering crust.
DOC concentrations in supraglacial stream water were lower than the DOC in all surface ice habitats, particularly
surface ice with high visible impurities (Fig. 5). This suggests a second possible mechanism of retention of DON
and DOP in the weathering crust, via the production of extracellular polymeric substances (EPS). Algae and bacteria
produce EPS which can alter the physical and chemical environment around their cells (Stibal et al.,
2012a;Angelaalincy et al., 2017). For example, it has been shown that EPS are used by cyanobacteria in cryoconite
holes to bind mineral particles together creating the cryoconite granules at the bottom of the hole (Stibal et al.,
2012b;Yallop et al., 2012;Musilova et al., 2016). EPS is often colliodal (here, operationally defined as passing
through 0.4 μm, but not 0.02 μmfilter membranes) (Raiswell et al., 2018), and when analysed from filtered (through
0.4 μm membranes), melted surface ice samples will be in the dissolved organic fraction (Pereira et al.,
2009;Hodson et al., 2010). The chemical composition of EPS exuded by glacier algae is unknown. We note that the
EPS of bacteria living in sewage sludge can have a molar C:N:P ratios that approaches 100:101:14 (Guibaud et al.,
2008), in order to illustrate that EPS can contain N and P. It is likely that the EPS of glacier algae contains relatively
more C than N and P, given the depauporate nature of the melting ice surface. EPS certainly seem to be associated
with the binding and retention of particulates in the weatheirng crust, and it follows at least some of the DON and
DOP may also be associated with this EPS.
These two mechanisms of retention of dissolved organic nutrient in the weathering crust, either in temporarily stored
water or as EPS, mean that DOC, DON and DOP storage in the weathering crust is transitory, and given the
dynamic response of the weathering crust to climatic perturbations, it is very likely that export of these species from
the weathering crust will be pulsed, rather than constant. For example, large melt events, accompanying summer
storms, may result in wholescale melting of the weathering crust (Tedstone et al., In Review), and export of a
significant quantities of the dissolved organic phases contained within them. By contrast, that stored in the
weathering crust towards the end of the ablation season, when the crust is freezing and water flow paths are closing,
may be retained in the frozen ice surface overwinter. For example, Musilova et al., (2017) reported that at the
margin of the GrIS, DOC remaining in surface ice at the end of the ablation season likely froze over winter and was
released the following ablation season through ice melt.

**4.4 Stoichiometry of different supraglacial environments**
DOC:DON:DOP ratios in melted surface ice samples may provide information on whether N or P is the limiting
nutrient within supraglacial environments in the Dark Zone. For example, Table 1 shows that DON:DOP ratios
increase systematically, from 49, 78 to 120, for low, medium and high impurity surface ice environments
respectively, as do DOC:DOP ratios (800, 1200, 2000). By contrast, DOC:DON ratios remain relatively stable for
the surface ice habitats (16, 16 and 17 respectively). This could indicate that P is limiting for the glacier algal
community. However, this does not quite tie in with the DIP data presented in Fig. 4, which shows that measurable,
if low, concentrations of P are usually present in the melting surface ice. Rather, $NO_3^-$ and $NO_2^-$ are below detection,
presumably as a result of uptake by phototrophs, and $NH_4^+$ is the only measurable DIN species, presumably as a
result of heterotrophic activity. Phototrophs preferentially utilize both $NH_4^+$ and $NO_3^-$, and the presence of both DIN
and DIP in melting surface ice environments, irrespective of visible particulate loading, and therefore of algal cell
abundance, suggests that a factor other than macronutrient concentration is limiting algal growth. Table 1 shows that
mean $NH_4^+$ concentrations in the melting surface ice are in the range of 0.6 – 1.0 μM. We noted above that there is
no readily available C:N ratio of glacier algae in the literature, but typical C:N ratios of sea ice algae are in the range
of 12-46 (Niemi and Michel, 2015). This implies that somewhere in the range of 7.2 – 26 μM of C could be
additionally fixed, if all the N was taken up by phototrophs with this range of C:N ratios. We also noted that it is
even more difficult to find C:N:P ratios of glacier algae, but should the C:P ratio be in the region of 100:1 to 1000:1,
then P demand will be 0.007 – 0.46 μM. Table 1 shows that the mean concentration of DIP in melting surface ice is
in the range of 0.03 to 0.05 μM, which suggests that P is not a limiting macronutrient on primary production. The
systematic change in DON:DOP and DOC:DOP ratios with increasing visible impurities, a proxy for algal cell
abundance, could be driven by the amount of P per cell that is potentially available at the high light intensity of the
ablation season (> 1500 μmol photons $m^2$ $s^{-1}$). The DIP content of the surface ice is relatively constant (Table 1)
given the much larger change in cell abundance as the visible impurities increase. The combination of lower P
availability at high light intensity results in an increase in the C:P ratio of phototrophs in other aquatic environments
(Hessen et al., 2013). It is plausible that this too happens with glacier algae, and that subsequent decomposition
products and EPS will likewise have higher DOC:DOP ratios as a consequence.

**5. Conclusion**
We conclude that DIN and DON concentrations in the melting surface ice of the Dark Zone on the GrIS are
markedly different from those documented in ice cores to date. Wolff et al., 2013 reported DIN, principally in the
form of $NO_3^-$, dominating the initial composition of ice melt, yet in the present study, DON dominates the melting
surface ice environments which host blooming glacier algae. Furthermore, DIN in these environments is exclusively
present as $NH_4^+$, and $NO_3^-$ is below the detection limit (0.64 μM). There is relatively little data on the P content of
Greenland ice, but we find that DOP dominates DIP in melting surface ice habitats, although DIP is usually present
in measurable quantities (LoD = 0.02 μM).   The presence of both $NH_4^+$ and DIP, even in heavily colonised melting
surface ice, suggests that factors other than macronutrient limitation control the blooms. We speculate that dissolved
macronutrients are held in the melting surface ice because of the architecture of the weathering crust, and/or because
EPS is retained within the melting ice latticework. The former controls the hydrology and the connectivity of water
flow paths and water storage in the surface ice, and the latter may be involved with the retention of particulates in
the surface. There is currently no data on C:N:P ratios of EPS exuded by glacier algae, but EPS of other autotrophs
does contain both N and P in association with C. DOC:DON ratios are relatively constant in melting surface ice, but
DOC:DOP ratios increase markedly with increasing algal cell counts. This may be attributable to the increasing cell
to DIP ratio, which, at high light intensity, increases the C:P ratio of autotrophs in other freshwater environments
(Hessen et al., 2013). This could be seen as a beneficial adaption to algal life in melting ice surfaces, where P
sources are limited, since blooms are not so dependent on P as a consequence.

## Data Availability

Holland, A., Williamson, C., Tranter, M., & Anesio, A. (2019). *Dissolved nutrient, carbon and algal abundance in*
*the Dark Zone (Greenland Ice Sheet), July-August 2016* (Version 1.0) [Data set]. UK Polar Data Centre, Natural
Environment Research Council, UK Research & Innovation. https://doi.org/10.5285/d8369a2f-8b50-4711-b492-
ae773bfafd95.

## Acknowledgments

The authors would like to thank and acknowledge the entire Black & Bloom team, especially those involved in the
sample collection conducted in the 2016 field season. The manuscript was considerably improved following the
constructive commentary of two anonymous reviewers.

## Team List

Liane G. Benning (GFZ German Research Centre for Geosciences, Potsdam, Germany), James B. McQuaid
(University of Leeds, Leeds, UK), Andrew J. Hodson (University Centre in Svalbard, Longyearbyen, Norway &
Western Norway University of Applied Sciences, Bergen, Norway), Edward Hanna (University of Lincoln, Lincoln,
UK), Tristram D. L. Irvine-Fynn (Aberystwyth University, Aberystwyth, UK), Jonathan L. Bamber (University of
Bristol, Bristol, UK), Stefanie Lutz (GFZ German Research Centre for Geosciences, Potsdam, Germany), Miranda J.
Nicholes (University of Bristol, Bristol, UK), Marek Stibal (Charles University, Prague, Czech Republic), Jason E.
Box (Geological Survey of Denmark and Greenland, Copenhagen, Denmark).

## Author contribution

MT, AA and MY conceived and designed the study. AH, CW, MT, AA, AT, JM, JC and the Black & Bloom group
collected the samples. CW provided algal counts for the mid to late ablation periods. AH conducted all the nutrient
analysis and was aided by FS in the instrument maintenance and data analysis. AH wrote the paper with inputs from
MT, CW, AT and AA. All authors reviewed the final manuscript.

**Competing Interests**
The authors declare they have no conflicts of interest.

**Funding**
This project has received funding from the European Union's Horizon 2020 research and innovation programme
under the Marie Sklodowska-Curie grant agreement No 675546. This work was also funded in part by the UK
Natural Environment Research Council Consortium Grant 'Black and Bloom' (NE/M0212025).
Figure 01. Map showing location of Camp BLACK & BLOOM 2016 (67°04'43.3"N, 49°20'29.7"W).
Background image sourced from Sentinel 2, taken on 26/7/2016.

Figure 02: The five supraglacial habitats sampled: **(a)** ice with low visible impurities, **(b)** ice with medium
visible impurities, **(c)** ice with high visible impurities, **(d)** supraglacial stream, **(e)** cryoconite hole.

Figure 03: Dissolved Organic Nitrogen (DON) and Dissolved Inorganic Nitrogen (DIN) concentrations for all
surface habitats (mean ± SE, n=19 for **L,M,H**, n=10 for **S** and n=14 for **C**). **L**- ice with low visible impurities,
**M**- ice with medium visible impurities, **H**- ice with high visible impurities, **S**- supraglacial stream water and **C**-
cryoconite hole water. LoD line depicts the limit of detection of the instrument. *Uppercase letters* denote
homogeneous subsets derived from post-hoc TukeyHSD analysis on a significant 1-way ANOVA in relation to
dissolved nitrogen phase. *Lowercase letters* denote T-test comparisons in relation to habitat type.

Figure 04:  Dissolved Organic Phosphorus (DOP) and Dissolved Inorganic Phosphorus (DIP) concentrations for
all surface ice habitats (mean ± SE, n=19 for **L,M,H**, n=10 for **S** and n=14 for **C**). **L**- ice with low visible
impurities, **M**- ice with medium visible impurities, **H**- ice with high visible impurities, **S**- supraglacial stream
water and **C**- cryoconite hole water. LoD line depicts the limit of detection of the instrument. *Lowercase letters*
denote T-test comparisons in relation to habitat type.

Figure 05: Dissolved Organic Carbon (DOC) concentrations for all five surface habitats (mean ± SE, n=19 for
**L,M,H**, n=10 for **S** and n=14 for **C**). **L**- ice with low visible impurities, **M**- ice with medium visible impurities,
**H**- ice with high visible impurities, **S**- supraglacial stream water and **C**- cryoconite hole water. LoD line depicts
the limit of detection of the instrument. *Uppercase letters* denote homogeneous subsets derived from post-hoc
TukeyHSD analysis on a significant 1-way ANOVA in relation to habitat type.

Figure 06: Algal cell abundance in ice surface ice habitats (mean ± SE, n=19 for each habitat). **L**- ice with low
visible impurities, **M**- ice with medium visible impurities and **H**- ice with high visible impurities. *Uppercase*
*letters* denote homogeneous subsets derived from post-hoc TukeyHSD analysis on a significant 1-way ANOVA
in relation to habitat type.

Figure 07: Conceptual diagram of the supraglacial environment in the Dark Zone of the GrIS. Black dashed
lines represent nutrient inputs to all supraglacial environments. Green lines represent hypothesized nutrient
inputs utilized by ice algal blooms. Arrow thickness represents relative nutrient concentration.
Table 01: Summary statistics for dissolved macronutrient (N and P) and DOC concentrations in the five
supraglacial habitats. DON, DIP, DOP and DOC denote Dissolved Organic Nitrogen, Dissolved Inorganic
Phosphorus, Dissolved Organic Phosphorus and Dissolved Organic Carbon respectively.
For each nutrient, the mean ± SD is provided, followed by the range of values. Concentrations are expressed in
µM; nutrient ratios are in µM/µM.

| | Ice Habitat | | | Supraglacial Stream | Cryoconite Hole | Field Blank |
|---|---|---|---|---|---|---|
| | Low | Medium | High | | | |
| $NH_4^+$ | 0.91±0.26 0-3.8 | 0.62±0.21 0-2.9 | 1.0±0.31 0-4.3 | 1.0±0.38 0-3.1 | 0.87±0.25 0-2.7 | 0.80±0.32 0-2.6 |
| $NO_2^-$ | 0.00±0.000 | 0.00±0.000 | 0.00±0.000 | 0.00±0.000 | 0.00±0.000 | 0.00±0.000 |
| $NO_3^-$ | 0.00±0.000 | 0.00±0.000 | 0.00±0.000 | 0.00±0.000 | 0.00±0.000 | 0.00±0.000 |
| DON | 5.1±1.1 0-10 | 11±2.0 0-40 | 14±1.7 3.2-27 | 0.00±0.00 0-0.82 | 0.70±0.32 0-3.2 | 0±0 0 |
| DIP | 0.03±0.02 0-0.27 | 0.07±0.02 0-0.44 | 0.05±0.01 0-0.20 | 0.01±0.01 0-0.04 | 0.06±0.02 0-0.23 | 0.00±0.00 0 |
| DOP | 0.10±0.02 0-0.27 | 0.15±0.02 0-0.48 | 0.12±0.01 0-0.25 | 0.07±0.03 0-29 | 0.07±0.02 0-0.22 | 0.00±0.00 0-0.04 |
| DOC | 83±24 0-350 | 173±30 29-451 | 242±44 0-636 | 30±13 0-84 | 50±33 0-435 | 12±7.7 0-35 |
| DON:DOP | 49.3 | 78.9 | 116.8 | 0.00 | 9.4 | Na |
| DOC:DOP | 797.8 | 1166.2 | 2013.3 | 455.3 | 671.3 | Na |
| DOC:DON | 16.2 | 15.6 | 17.2 | Na | 71.3 | Na |
| DIN:DIP | 27.2 | 8.4 | 19.6 | 74.1 | 15.5 | Na |
| Sample Size (n) | 19 | 19 | 19 | 10 | 14 | 9 |

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
