# Peer review of "Nutrient cycling in supraglacial environments of the Dark Zone of the Greenland Ice Sheet"

_Biogeosciences, 2019_

## Referee Comment (RC1) · Anonymous Referee #1 · 15 Apr 2019

GENERAL COMMENTS: Overall, the manuscript by Holland et al. provides some important, hard-fought observations in one of Earth's least studied biomes, and provide some of the first evidence of the biogeochemical role played by the large seasonal algal bloom that develops on Greenland's Ice Sheet, which has recently attracted attention due to its influence on albedo. These data are therefore timely given the projected future mass loss of the Greenland Ice Sheet, and the consequences that these fluxes may have on downstream environments. Lastly, these present data are unique in that they seem to represent a relatively large spatial and temporal extent, and analytically, the methods employed for the data generation appear excellent.

However, I have some concerns with the way the data are described, interpreted, and reported. Firstly, I feel like the authors could do better job in focusing what exactly this

paper is about, as the abstract, introduction, and discussion all give slightly different objectives for the study (see detailed comments below). I think that this manuscript would benefit from clarifying and focusing the objectives and hypotheses, and making these consistent throughout the document.

The second issue is in reference to the biogeochemical cycles/transformations hypothesized to be taking place on the surface of the icesheet. Some of the language in this regard could be tightened for accuracy and consistency (or at least clarified, see below comments), and I have suggested that the authors could create a conceptual diagram (with all inputs, outputs, transformations, etc) to help in presenting the hypotheses and afterwards discuss the data.

Thus, in revising this article, I challenge the authors to focus this research by asking specific, testable questions, and clearly using the data to answer these questions throughout the different sections of the document, as well as to pay careful attention to the biogeochemical transformations taking place in this special environment. Some specific comments are outlined below by section and line number.

SPECIFIC COMMENTS Title: Is the paper really about nutrient 'cycling'? Maybe something like 'organic nutrients dominate supraglacial environments and correlate with algal cell density...' or similar would better represent the subject matter of this paper.

ABSTRACT Line 19: Probably should be nutrient 'abundance' rather than nutrient 'cycling' that is a constraint on algal abundance. Also, do we know if nutrients are indeed a constraint on these communities?

Line 20: This paper does not really investigate the conversion of dissolved inorganic nutrients to organic ones; it more just investigates the abundance of each. We can of course infer that conversion is the reason for one form of nutrient over another, but most certainly conversion itself was not assessed.

Lines 21-22: Where are these percentages coming from....are these from the entire

dataset? There was a gradient of algal abundance sampled over, as well as cryoconite and supraglacial stream categories. . ..it might be appropriate to describe the sampling scheme briefly in the abstract, and state which of these data were used to calculate these numbers.

Line 23-24: Can maybe be more specific here to indicate the shift from inorganic to organic forms rather than 'phase shift'.

Line 24-25: Again, what supraglacial environments are we referring to with these ratios? There are three values given after DON:DOP and DOC:DOP. . ..why three - what do they correspond to? Also, why were these ratios reported and not DOC:DON? Perhaps more importantly, why are only the organic forms being reported and compared with Redfield Ratio as opposed to inorganic forms?

INTRODUCTION Line 40 and 56: Particles of what? Given the potential importance of these particles in providing nutrients, I think they can be described in a bit more detail here. Are these the same particles described in lines 41-44 as being LAI's?

Line 60: Redfield et al., 1963 is an interesting choice for a reference, especially since it is regarded as being specific only to marine plankton in the discussion. Could maybe find something more broad and recent. . ..maybe the Ecological Stoichiometry book by Sterner and Elser (2002) would work better?

Line 60: Why is carbon in ready supply on the ice sheet surface; where is it coming from? Why would this not also be the case for nitrogen and phosphorus. . ...where are these two coming from and in what forms? Perhaps this is intuitive to the authors who are specialists for this ecosystem type, but would be good to describe some of these inputs/outputs to non-specialist readers of the journal.

Line 63: Does the 'Stibal et al. 2017a' citation go with the cell concentration number? If so, it might be better to move it there. . ...I'm not sure that paper suggests that these habitats are nutrient rich (but I could be wrong).

Line 69: If there are some more examples than the Telling et al. 2012 paper, you should cite them here.

Line 71: What was the detection limit in this study (i.e. Telling et al. 2012)? Should report before the citation in the same units as your paper.

Line 73-75: This is more or less what you found for DIN, no? However, for DON, the values were much greater. I think it would be nice to revisit these ideas in the discussion.

Line 76-77: This sentence is a little confusing to me…..how do cycles of uptake and remineralization lead to accumulation of nutrients in biomass? Also, I think there are potentially a lot of systems with microbially-mediated nutrient cycles that can be used as an analogue here….Planktonic aquatic systems are nice ones, but I don't think this is somehow the pinnacle of nutrient cycling.

Line 78: Maybe rephrase this…..'extremely active nutrient cycling' sounds strange and unspecific to me. Would be better to give a rate estimate.

Line 79: I think this is something that you need to expand a bit more on, since the whole paper is essentially centered on it. Why are dissolved nutrients concentrating in the organic form, and is this really a sign of 'active' nutrient cycling? Later in the text, the opposite rationale is essentially used to explain the same observation, which is that low mineralization rates are responsible for an accumulation of organic nutrients. I think the authors would do well to describe the major inputs, outputs, and transformations in this unique habitat. Perhaps a conceptual diagram could help here, not only explain the rationale for this nutrient survey, but also help define your hypotheses/predictions?

Line 80-82: Isnt there organic nutrient data in Telling et al. 2012? It is likely that there are not so many reports of organic nutrients from the dark zone of the GRIS (its not so easy to get there, afterall), but what about elsewhere on the ice sheet, or on other glaciers around the world? I think this is something, in concert with my comment above,

that needs to be expanded upon ultimately given the content of this paper, an in order to appreciate the finding of this paper later.

Line 84: Do you expect that the ice algae are 'recycling' the nutrients, or just taking them up?

Line 88: I think you would need uptake data, for example, to actually evaluate the 'importance' of different nutrient forms. Also, when you say 'microbial' recycling, are you only talking about the algae?

METHODS Line 98: This is an extremely big area. How were sites randomly sampled (line 103) over such a large patch? Is there any sense of the area covered/sampled over this time? Were some sites/areas resampled over the month of fieldwork?

Line 99: Was there any relationship with nutrient concentrations and date sampled? I can imagine that conditions on the ice could be a lot different on the 15 of July than they are on the 15 of August.

Line 100: This explains why you sampled the surface ice in low, medium, and high categories, but did you really sample the cryoconite and streams due to the spatial heterogeneity in ice algae distribution? Algae were not quantified for these two habitats, so this is probably not the case. If it is just as a comparison with the surface ice that is fine, but some justification is warranted.

Lines 109-110: Was there any special preparation for the glass stack, bottles and collection jars? Eg. Acid washing, furnacing, etc?

Line 131: What was the purpose in assessing the assemblage diversity (as opposed to just a number of cells)?

Line 140: What is TON....total oxidized nitrogen? Should probably spell this out the first time.

Line 143: This is a bit confusing as written...why not say that DON was estimated by

subtracting DIN from TDN since you already defined DIN above? Or would be easier to say DON=TDN-DIN?

Line 166: Why cite RStudio here...Wouldn't it better to cite R?

Lines 166-172: In general, I think that it would be better to be more specific about what analyses were conducted and why. For example, can say in order to test hypothesis 'x', we performed test 'y'.

Line 170: Similar to the comment above, why test DON and DOC, but not DOP? Why were these parameters chosen, and how to they help you to achieve your objectives? For example, why would you not look at inorganic species, or the ratio of organic to inorganic forms as a function of cell abundance? Would it help to include sample date and spatial coordinates as random variables?

General comment: Was there any attempt to quantify particulates on the surface ice? While biological activity is no doubt important to biogeochemical cycling, so too would be the density of particulates I would think, especially with regard to phosphorus, since it is usually sediment-bound. While this paper of course focuses on the dissolved fraction, the particulate fraction is likely also important, and I feel like this would also help answer a similarly important question: are the nutrients in the forms they are because of the biological actors, or because of what the biological actors are sitting upon? This may also play a role in why some patches are in 'high abundance', and others are in 'low abundance', and thus would be collinear with cell abundance. Also, if a given sample was below detection, were they included in the analyses? They seem to be included in the figures, but would be good to know if they were also included in calculations, and if so which ones and how they were treated?

RESULTS General comment: I think it would make more sense if the results section was more hypothesis-lead as well. Right now, it reads more like a list with some carefully chosen significant relationships scattered about, and are difficult to understand how they relate to the overall picture.

Line 176: In some ways, I feel like this opening sentence is really only validating the obvious. Transects were chosen based on the abundance of stuff covering them, and the first result is that more stuff was found in these patches covered with more stuff. I think that it would be more helpful to report it in this way such that it is setting up your experimental design rather than a unique result in its own right.

Line 179-182: Why are correlations with DOC and DON reported here and not below? Why did you not compare with DOP? Also, while an interesting result, I feel like calling them 'highly significant' is a bit excessive, since the relationships (as far as I can tell anyway) seem to be based upon 3 comparisons apiece (averages of low, mid, high). Would Pearson correlations be the correct test here, or would it be better to test against the categories?

Line 184 and elsewhere: Noting the number of samples that were over the LOD is great, but out of how many samples? What then happens to these below detection numbers….are they included in calculations? Also, are some of these replicates or from the same patches? Are these also including cryoconites and supraglacial streams? The authors need to be more specific in their reporting of these data.

Line 186: This is interesting….why do you think that $NH_4$ was the dominant component of the DIN? Could this be from microbial ammonification of DON? I think this could be potentially also highlighted in a conceptual diagram!

Line 194-200: Again, why do you not make comparisons with abundance and DOP? it seems central to what you are trying to find out, whether or not comparisons are 'significant' (in either case its interesting). It is also not clear which samples you are talking about ….are they all pooled values for the icesheet as a whole?

Line 196: 'The mean concentrations for the remaining 40 DIP concentrations [that were above the LOD] ranged from 0-0.7'….the lower limit should be 0.02, since that was the limit of detection, right?
Line 198: 'DOP concentrations in cryoconite hole and supraglacial stream water fell below the LOD'....How do you mean this...that they fell below the LOD sometimes? In Figure 5, the average DOP for these two habitat types is around 7 uM. DON is a different story.....Could it be that these two are being confused?

General comment: There are several mentions of nutrient ratios in the abstract and discussion. Why are these not discussed in the results? Also, where is figure 7?

DISCUSSION Lines 212-214: This information should be in the results, and it should be specified how they are calculated. For example, are these calculated for only surface ice environments? Furthermore, I think that the ratios of organic to inorganic nutrients would be potentially equally or more interesting to correlate with algal cell abundance than the absolute concentrations.

Line 215: Has this dominance been reported in other glacial systems?

Line 222: Does Tedstone et al. 2017 actually report the timing of this shift in Nitrogen? Actually, has anyone reported this shift in nitrogen?

Lines 223-225: Similarly, how does this Williamson et al. (2018) paper support the shift in nitrogen phase? I think this needs to be rephrased/recast.

Lines 226: But, these other impurities were not quantified, so its difficult to say this for certain....For all we know, all the impurities could be ice algae! However, I think there may be some other papers showing this these days that you can cite...

Line 227: There is a lot of talk of nutrients 'shifting' to the organic phase. But, it looks like to me that the concentration of DIN is basically the same for all the surface ice habitat types. Might the DON rather be accumulating through time from ice algae taking up DIN and subsequently 'leaking' DON into their habitat, rather than the DIN pool shifting? It would really be nice to see these relationships over time.

Line 228: Furthermore, the big differences in organic/inorganic nutrients with algal biomass seems to only apply to nitrogen, and I think it is important that this distinction

is made. Why would this not apply to phosphorus? This should be discussed in detail, and the authors should be more specific whether they are talking about 'nitrogen' or indeed 'nutrients' (ie nitrogen + phosphorus) elsewhere in the manuscript.

Line 230: Do the data really suggest 'efficient' conversion? I think at best there is a correlation between cell counts and organic nutrients, but no data that points directly to conversion, and definitely no data that would suggest that the process is efficient (for example, the DIN concentration seems unchanged with increasing cell abundance). Furthermore, why do you think the same would not be seen for DOP?

Line 232-233: I think this information belongs in the results section. Furthermore, Figure 7 is mentioned for the first time here. Maybe would it be better to put this in supplementary information if it is not going to be used to support the main results? Individual data points could also be superimposed onto bar figures (e.g. 'jittered' points in ggplot2) to illustrate variability between categories, if that is the goal.

Line 239: 'Demonstrate' is strong in this case....perhaps 'suggests'?

Line 240-241: Are ice algae assemblages the main producers of dissolved organic nutrients stocks in freshwater and marine ecosystems? Recast this text.

Line 242: Do the ice algae really 'rapidly' take up inorganic nutrients? If there are some numbers to back this statement that is great, but I think this cannot be said without some support.

Line 243: I still think that it would help to somehow organize these sources in a diagram to help guide your thinking and the readers comprehension. What forms of inorganic nitrogen is deposited on the ice sheet and how? How about organic forms? Phosphorus?

Line 245: Can also be breakage, leakage, or lysis, for example....what about extracellular processes?

Line 248: Does bacterial carbon production equate to nutrient-transformation processes like ammonification? If bacteria are really that sparse, I think you could alternatively think that they are really efficient, since they seem to be producing measurable ammonium in excess of uptake.

Line 251-254: 'Reduced capacity' is interesting wording. . ..were they at higher capacity at some point? I think the production of ON is just outpacing mineralization

Lines 257-259: This is interesting that all of these different habitats types studied by Stibal et al. (2008) also had the organic forms dominate. Why do you think this was not the case for Nitrogen in the supraglacial streams and cryoconites from this study, while it also it seems to hold true for phosphorus?

Line 271: Are ice algae producing EPS? Has anyone tried to quantify this?

Line 279-280: is it possible that DON and DOP are also 'over-wintering' on top of the icesheet? Could any of this be 'leftovers' from a previous season?

Line 280: This sentence is vague. . ..what exactly about the export of dark zone DOM is unknown. . ..the character. . .the quantity?

Line 285: The Redfield Ratios was certainly generated using data from marine systems, but I think its utility over the last decades has been in providing a point of comparison. However, I think it also deserves clarification that the Redfield Ratio is the average molar ratio of biomass under balanced growth. Do we know the elemental composition of ice algae under balanced growth, and how it compares to Redfield Ratio? I'm also not sure that I understand the purpose of the text that follows. While there is certainly a lot of variability across aquatic habitats in dissolved N:P ratios from cold regions around the world (and elsewhere), I'm not sure how useful it is to bring up these numbers here. Furthermore, it is not clear if the ratios from the cited studies are also using the organic fractions-only as done in this study (my guess is that this is not the case). If the purpose of this text was to (presumably) link the reported N:P ratios discussed in the paragraph below to the literature, this makes comparisons difficult,

and calls into question the need for this text, or at least would suggest that it needs to be revised to fit the authors' purpose.

Line 295: This is the first time DOC:DON:DOP ratios have been reported besides in the abstract. . ..I did not see it in the introduction, methods, or results that you planned to look at these ratios.

Lines 298-300: Why are you making nutrient ratios for the organic form of these nutrients? Wouldn't you expect that algae would be taking up the inorganic forms primarily (especially NH4)? I feel like these ratios might not be accurately approximating availability for algae, and thus I'm not sure that, based on comparing these ratios with the Redfield Ratio alone, that we can say that the system is P-limited. I think it needs to be carefully explained in the text why this would be the case.

Lines 301-304: Would it be possible to more rigorously investigate this statement of different slopes of CP and CN over algal abundance? I think that this could be interesting if better developed, but as written it seems more of an afterthought.

Line 313: Is cryoconite the same as the particles talked about in the introduction?

Lines 326: In order to be able to say 'rapid uptake of dissolved nutrients', you need to have data on the uptake rates to compare. You also do not report rates of organic production.

Lines 328-329: These production data are also assumed to hold true here, as production wasn't investigated in this work. Also, why would it be inefficient. . ..because there are leftover organic nutrients?

Line 332: Was this the case for phosphorus? Also, I think that the notion of this retention being due to EPS is too speculative to say it this way.

Line 334: This is vague and repeated from line 280.

TECHNICAL CORRECTIONS Line 23: Comma after 'nitrogen' not necessary

Line 30: Should there be spaces between values and "Gt"?

Line 36: Similarly, should there be a space between "30" and "km"? This should be fixed throughout.

Line 160: HCl

Line 214: comma after "To date"

Line 241: here and elsewhere, references should be ordered.

---

## Referee Comment (RC2) · Anonymous Referee #2 · 29 Apr 2019

This paper provides novel information on the chemistry of supraglacial ecosystems. The main finding is that most of the dissolved N and P in these environments is in organic rather than inorganic forms. The authors use their chemical data in concert with measurements of algal cell abundance to make inferences about the role of microbes in supraglacial nutrient cycling. The paper is generally well written and would be of interest to biogeochemists, and to a lesser extent, hydrologists and glaciologists, working in ice-covered ecosystems. There are several sections of the paper that I felt were overly speculative, especially with regard to rates and mechanisms of nutrient retention. In addition, I believe that the authors could better reconcile their findings with previous literature on OM production in supraglacial environments. As a result, I think the paper needs some important revisions before it should be considered for

publication in Biogeosciences Discussions. I have provided comments and editorial suggestions below that I hope will be helpful for revising the paper.

General Comments

Line 99: It would be appropriate to report the number of samples collected for each habitat type somewhere in this section.

Line 101: How were sample locations classified into low, medium, and high impurity categories? The figure gives a sense of the density of impurities but there is no indication of whether there was some quantitative aspect to the process (i.e. number of impurities per unit area) or whether the process was wholly subjective. Also, the nature of the impurities is not well described – are they mineral, biological, or a mixture of both (such as the material found in cryoconite holes)?

Lines 179-181: The comparisons between algal cell abundance and organic nutrients are inconsistent. Algal cells and DOC are compared by regression, algal cells and DOC and DON are compared by pearson correlation, and algal cells are not compared at all to DOP. Moreover, these tests do not provide any information about the differences in the relationship between different habitats.

Line 184: What was the LoD for DON? Are the sample numbers you report (54 DON samples, 41 DIN samples) out of the 70 samples you included in the data for Figure 4? Also, what value did you use for all of the samples that were below the LoD – half of the LoD or some other value?

Line 194: What was the LoD for DOP? There were 74 DOP samples above the LoD, however in the legend for Figure 5 it appears that only 70 DOP samples were included in the figure.

Line 230: How do you get information about conversion rates from the concentrations you measured?

Lines 232-234: This regression plot is not an effective way to analyze the relationship

(or lack thereof) between DOC and algal abundance. The fact that there is any positive relationship is based on the single outlier in the upper right hand corner of the graph. If you removed that outlier, it appears that there would be a negative relationship between DOC and algal cell abundance (or, at best, no relationship). If there is, in fact, no relationship between algal cell abundance and DOC, that does not seem to support your statement that you "interpret these data to demonstrate that ice algal assemblages are the main producers of dissolved organic nutrient stocks within the melts surface ice..." (line 239). This may well be true but it is not what these data show. There are other possible explanations for the lack of relationship between DOC and algal cell counts including that you are comparing data collected across a full month and the relationship may change over the melt season.

Lines 234-237: Similar to the comment above, this explanation for the lack of a relationship between algal cell abundance and DOC would be more convincing if it detailed more specifically how these variables could become decoupled rather than just invoking the "highly dynamic nature of the environment" where solutes and gases move around.

Lines 267-274: It is surprising that cryoconite holes have low stocks of dissolved organic nutrients compared to surface ice. Past research has focused on cryoconite holes as hotspots of C fixation in autotrophic supraglacial environments (e.g. Anesio et al., 2009, Global Change Biology). If this were the case, it seems that the abundant production in cryoconite holes would be reflected in dissolved organic nutrient concentrations, but that is not what these data show. Does this suggest that surface ice habitats are potentially more important for autotrophic production or is there another explanation? Also, if you invoke EPS, which is known to occur in cryoconite holes, as the mechanism by which nutrients are retained in surface ice, wouldn't this also be true for cryoconite holes and drive up dissolved organic nutrient concentrations in the same way in those habitats?

Line 277: I don't find the argument for a "large pulse of dissolved organic nutrients" particularly convincing. Particulate organic nutrients are hardly mentioned in this paper. It seems like a more parsimonious explanation for the loss of organic nutrients produced in supraglacial habitats is that they are exported downstream, at least partially, in particulate forms.

Editorial Suggestions

Line 77: change "to accumulate" to "accumulation"

Line 101: add "of" after "amounts"

Line 140: It would be helpful to define the acronym TON. I presume that it represents total oxidized nitrogen here but this acronym is commonly used to refer to total organic nitrogen (dissolved + particulate ON) so you should be clear about how it is being used.

Lines 179-180: This sentence refers to data shown in Fig. 7 (currently referenced on line 233), which should be renumbered to Fig. 4 and cited here.

Lines 189: "increase" should be "increased" to be consistent with the rest of the results which are in the past tense.

Line 234: Suggest changing "counts were" to "abundance was" since DOC is not plural.

---

## Author Response (AR1)

**Author's Responses to Reviewer 1:**

**Overall response:** We would like to thank the reviewer for the helpful and constructive review. We have made extensive changes to the text, particularly the Discussion, in line with the commentary below and that of the other Reviewer. We feel that the manuscript has been significantly improved as a consequence.

**Reviewer comments and responses.**

GENERAL COMMENTS: Overall, the manuscript by Holland et al. provides some important, hardfought observations in one of Earth's least studied biomes, and provide some of the first evidence of the biogeochemical role played by the large seasonal algal bloom that develops on Greenland's Ice Sheet, which has recently attracted attention due to its influence on albedo. These data are therefore timely given the projected future mass loss of the Greenland Ice Sheet, and the consequences that these fluxes may have on downstream environments. Lastly, these present data are unique in that they seem to represent a relatively large spatial and temporal extent, and analytically, the methods employed for the data generation appear excellent.

However, I have some concerns with the way the data are described, interpreted, and reported. Firstly, I feel like the authors could do better job in focusing what exactly this paper is about, as the abstract, introduction, and discussion all give slightly different objectives for the study (see detailed comments below). I think that this manuscript would benefit from clarifying and focusing the objectives and hypotheses and making these consistent throughout the document.

• **Response:** We would like to thank the reviewer for their commentary regarding the clarification of the manuscript. We have significantly revised the manuscript and feel that this revision better follows the three main aims and objectives set out in the last paragraph of the introduction. A major restructuring of the results section has been carried out, to make the section more hypothesis driven, as well as to link more clearly to the objectives in the introduction and the subsections of the discussion. The discussion has also been rewritten to clarify meaning and refocus on the objectives of the manuscript.

The second issue is in reference to the biogeochemical cycles/transformations hypothesized to be taking place on the surface of the ice sheet. Some of the language in this regard could be tightened for accuracy and consistency (or at least clarified, see below comments), and I have suggested that the authors could create a conceptual diagram (with all inputs, outputs, transformations, etc) to help in presenting the hypotheses and afterwards discuss the data.

• **Response:** We hope that we have removed terms and phrases that could be misleading to the reviewer about what was actually quantified in the present study. A simple conceptual diagram has been included.

Thus, in revising this article, I challenge the authors to focus this research by asking specific, testable questions, and clearly using the data to answer these questions throughout the different sections of the document, as well as to pay careful attention to the biogeochemical transformations taking place in this special environment. Some specific comments are outlined below by section and line number.

• **General response:** We thank the reviewer for this challenge and hope that the revision now passes muster. Our responses to each individual question are given below.

Title: Is the paper really about nutrient 'cycling'? Maybe something like 'organic nutrients dominate supraglacial environments and correlate with algal cell density...' or similar would better represent the subject matter of this paper.

• **Response:** We agree and have changed the title.

ABSTRACT Line 19: Probably should be nutrient 'abundance' rather than nutrient 'cycling' that is a constraint on algal abundance. Also, do we know if nutrients are indeed a constraint on these communities?

• **Response:** the text has been revised. Nutrient abundance in the Dark Zone has not been investigated in detail enough to definitively determine if it is a constraining factor on the bloom, which is why this manuscript investigates a limiting nutrient.

Line 20: This paper does not really investigate the conversion of dissolved inorganic nutrients to organic ones; it more just investigates the abundance of each. We can of course infer that conversion is the reason for one form of nutrient over another, but most certainly conversion itself was not assessed.

• **Response:** Text changed from 'conversion of dissolved inorganic nutrients...' to 'abundance of dissolved organic nutrients....'.

Lines 21-22: Where are these percentages coming from. . .are these from the entire dataset? There was a gradient of algal abundance sampled over, as well as cryoconite and supraglacial stream categories. . ..it might be appropriate to describe the sampling scheme briefly in the abstract, and state which of these data were used to calculate these numbers.

• **Response:** The authors have added text briefly describing the five supraglacial environments sampled in the study. Please see lines 20-21 of updated manuscript. Text has been added explaining that the percentages have been calculated from across all of the ice surface samples containing low, medium and high visible impurity loadings. Please see lines 21-22 of updated manuscript.

Line 23-24: Can maybe be more specific here to indicate the shift from inorganic to organic forms rather than 'phase shift'.

• **Response:** This line has been deleted in updated manuscript.

Line 24-25: Again, what supraglacial environments are we referring to with these ratios? There are three values given after DON:DOP and DOC:DOP. . ..why three - what do they correspond to? Also, why were these ratios reported and not DOC:DON? Perhaps more importantly, why are only the organic forms being reported and compared with Redfield Ratio as opposed to inorganic forms?

• **Response:** This line has been deleted in updated manuscript.

INTRODUCTION Line 40 and 56: Particles of what? Given the potential importance of these particles in providing nutrients, I think they can be described in a bit more detail here. Are these the same particles described in lines 41-44 as being LAI's?

• **Response:** The particles being referred to are considered to be dust, dated to the late Holocene by Wientjes et al., 2012, melting out of ancient meteoric ice. However, these particles are one example of mineralogic LAIs that could comprise the visible impurities seen in the Dark Zone, which is why the authors have also included a list of other mineralogic LAIs in Lines 53-54 of the updated manuscript. In line 52 of the updated manuscript 'ancient Holocene dust' has been added as a descriptor of the particles being described.

Line 60: Redfield et al., 1963 is an interesting choice for a reference, especially since it is regarded as being specific only to marine plankton in the discussion. Could maybe find something more broad and recent. . .maybe the Ecological Stoichiometry book by Sterner and Elser (2002) would work better?

• **Response:** We have largely removed reference to the Redfield ratio, and note that information on the C:N:P ratio of glacier algae is sparse. We have added Hessen et al., 2013 as a additional reference.

Line 60: Why is carbon in ready supply on the ice sheet surface; where is it coming from? Why would this not also be the case for nitrogen and phosphorus. . .where are these two coming from and in what forms? Perhaps this is intuitive to the authors who are specialists for this ecosystem type but would be good to describe some of these inputs/outputs to non-specialist readers of the journal.

**Response:** Carbon is in ready supply on the ice sheet surface for two main reasons, the first is • that it is scavenged from the atmosphere during snow crystal formation and then is released to the surface ice environments when the snow pack ablates. The second is due to the surface ice environments constant interaction with the atmosphere. Due to the air-water interface during the main ablation season, gas exchange can occur which allows for carbon to be readily available. Both of these forms of carbon are in the dissolved inorganic phase, which includes aqueous CO2, HCO3 (bicarbonate), and H2CO3 (carbonic acid). Nitrogen is dominantly released to supraglacial environments via snow melt as nitrogen is also scavenged from the atmosphere during snow formation, with a lesser input from ice ablation. N2 is also a potential source due to the air-water interactions occurring as mentioned before, however it is not very bioavailable and most photosynthetic organisms are not able to fix it from the atmosphere (Falkowski and Raven, 1997). Furthermore, Telling et al., 2012 reported that the overall importance of nitrogen fixation for microbial growth decreases with distance from the margin of the GrIS. Phosphorus is a rock dervived and is therefore only released by physical and chemical weathering of rock derived particles. Typically why it is the limiting nutrient in supraglacial environments. Lines 69-78 have been updated to include a more detailed explination.

Line 63: Does the 'Stibal et al. 2017a' citation go with the cell concentration number? If so, it might be better to move it there. . .I'm not sure that paper suggests that these habitats are nutrient rich (but I could be wrong).

• **Response:** This line has been deleted in updated manuscript.

Line 69: If there are some more examples than the Telling et al. 2012 paper, you should cite them here.

• **Response:** References to Telling et al., 2012 and Wadham et al., 2016 has been added to the end of lines 85-86 of updated manuscript as these presently are the only two studies to have quantified nitrogen concentrations in the Dark Zone of the GrIS.

Line 71: What was the detection limit in this study (i.e. Telling et al. 2012)? Should report before the citation in the same units as your paper.

• **Response:** The LoD for the Telling et al., 2012 study was 0.33µM and has been added to line 89 of the updated manuscript.

Line 73-75: This is more or less what you found for DIN, no? However, for DON, the values were much greater. I think it would be nice to revisit these ideas in the discussion.

• **Response:** Section 4.1 of the discussion revisits these values and discusses the difference between the DIN and DON concentrations for this study.

Line 76-77: This sentence is a little confusing to me. . .how do cycles of uptake and remineralization lead to accumulation of nutrients in biomass? Also, I think there are potentially a lot of systems with microbially-mediated nutrient cycles that can be used as an analogue here. . .Planktonic aquatic systems are nice ones, but I don't think this is somehow the pinnacle of nutrient cycling.

• **Response:** Uptake and remineralization describes what occurs in the microbial loop. As these microorganisms are utilizing and recycling the available nutrients, they become incorporated and accumulate into their cellular biomass as well as to being released back into the meltwater, which leads to nutrients not only existing in the environment in the inorganic phase but also in the organic phase in the form of biotic mass and dissolved organic matter that the cells produce. The authors chose to use planktonic aquatic systems as a comparison as it is similar to the aqueous ice surface environments.

Line 78: Maybe rephrase this. . .'extremely active nutrient cycling' sounds strange and unspecific to me. Would be better to give a rate estimate.

• **Response:** Line 95 of the updated manuscript now cites an NPP rate from Williamson et al., 2018.

Line 79: I think this is something that you need to expand a bit more on, since the whole paper is essentially centered on it. Why are dissolved nutrients concentrating in the organic form, and is this really a sign of 'active' nutrient cycling? Later in the text, the opposite rationale is essentially used to explain the same observation, which is that low mineralization rates are responsible for an accumulation of organic nutrients. I think the authors would do well to describe the major inputs, outputs, and transformations in this unique habitat. Perhaps a conceptual diagram could help here, not only explain the rationale for this nutrient survey, but also help define your hypotheses/predictions?

• **Response:** We argue that DIN and DIP uptake by glacier algae and the production of EPS and other degradation products is the source of DOP and DON. This is the most consistent explanation from the data sets we present. A simple conceptual diagram is now included in the manuscript.

Line 80-82: Isn't there organic nutrient data in Telling et al. 2012? It is likely that there are not so many reports of organic nutrients from the dark zone of the GRIS (it's not so easy to get there, afterall), but what about elsewhere on the ice sheet, or on other glaciers around the world? I think this is something, in concert with my comment above, that needs to be expanded upon ultimately given the content of this paper, an in order to appreciate the finding of this paper later.

• **Response:** Text has been added that now cites Telling et al., 2012 and Wadham et al., 2016 as reporting TN for the Dark Zone, and clarifying that non has been reported for ice populated by Streptophyte ice algae, lines 111-112. Lines 104-107 cite other sources reporting dissolved organic nutrient concentrations in other Arctic environments and the Antarctic.

Line 84: Do you expect that the ice algae are 'recycling' the nutrients, or just taking them up?

• **Response:** This line has been deleted due to edits in the updated manuscript. We do believe that recycling is occurring due to the fact that heterotrophs are present. If heterotrophs are present then they are utilizing dissolved organic matter and therefore remineralization is occurring, albeit at an inefficient rate. Lines 102-118 begin to describe this conclusion.

Line 88: I think you would need uptake data, for example, to actually evaluate the 'importance' of different nutrient forms. Also, when you say 'microbial' recycling, are you only talking about the algae?

• **Response:** Text changed from 'importance' to 'relative abundance' in response to reviewer's comment, line 129 in updated manuscript. 'Microbial' recycling refers to the microbial loop and therefore both the algae and bacteria.

METHODS Line 98: This is an extremely big area. How were sites randomly sampled (line 103) over such a large patch? Is there any sense of the area covered/sampled over this time? Were some sites/areas resampled over the month of fieldwork?

• **Response:** Within each category of low, medium and high visible impurity loadings the sample location was chosen randomly by eye. There was no quantification of visible impurity loading before sampling however, as seen in Figure 2 the differences between the three ice surfaces are very apparent. Figure 3 reinforces this by the significant difference in algal abundance between the three ice surfaces. GPS points were collected at each sampling location within the 500 X 500 m sampling site, however no plots have been made to visualize the total area covered. Sampling areas were destructively sampled by the use of a hand saw to remove the top 2cm of the surface ice as described in the methods, therefore areas were not resampled.

Line 99: Was there any relationship with nutrient concentrations and date sampled? I can imagine that conditions on the ice could be a lot different on the 15 of July than they are on the 15 of August.

• **Response:** There was no clear temporal trend in the data. We believe that this is due to the extremely dynamic and heterogenic nature of these environments making trends over long time series difficult.

Line 100: This explains why you sampled the surface ice in low, medium, and high categories, but did you really sample the cryoconite and streams due to the spatial heterogeneity in ice algae distribution? Algae were not quantified for these two habitats, so this is probably not the case. If it is just as a comparison with the surface ice that is fine, but some justification is warranted.

• **Response:** Lines 144-145 in the updated manuscript adds text for clarification about supraglacial stream and cryoconite hole sampling.

Lines 109-110: Was there any special preparation for the glass stack, bottles and collection jars? Eg. Acid washing, furnacing, etc?

• **Response:** Lines 158-160 in the updated manuscript adds text explaining the sample jar preparation.

Line 131: What was the purpose in assessing the assemblage diversity (as opposed to just a number of cells)?

• **Response:** Reference to assemblage diversity has been removed.

Line 140: What is TON. . ..total oxidized nitrogen? Should probably spell this out the first time.

• **Response:** Line 185 does this.

Line 143: This is a bit confusing as written. . .why not say that DON was estimated by subtracting DIN from TDN since you already defined DIN above? Or would be easier to say DON=TDN-DIN?

• **Response:** Text has been changed for clarification, line 193.

Line 166: Why cite RStudio here. . .Wouldn't it better to cite R?

• **Response:** RStudio is considered an IDE, integrated development environment which cannot be run without R. However, R is an independent program, which can be run without RStudio.

Lines 166-172: In general, I think that it would be better to be more specific about what analyses were conducted and why. For example, can say in order to test hypothesis 'x', we performed test 'y'.

• **Response:** We feel the text does say this. Please push back if you still feel that it doesn't.

Line 170: Similar to the comment above, why test DON and DOC, but not DOP? Why were these parameters chosen, and how to they help you to achieve your objectives? For example, why would you not look at inorganic species, or the ratio of organic to inorganic forms as a function of cell abundance? Would it help to include sample date and spatial coordinates as random variables?

• **Response:** Line 250 of the updated manuscript now state: 'Comparison of DOP surface ice concentrations and algal cell counts were not significant.'. Only significant relationships were reported in the manuscript. The authors chose to compare the average algal abundance to the average DOC and DON concentrations for the three ice surfaces as a way of illustrating a relationship between the glacier algae abundance and the concentration of DOC/DON. We feel that this helps achieve the objective of showing that algae are the main producers of DOC and DON in the ice surface environments.

General comment: Was there any attempt to quantify particulates on the surface ice? While biological activity is no doubt important to biogeochemical cycling, so too would be the density of particulates I would think, especially with regard to phosphorus, since it is usually sediment-bound. While this paper of

course focuses on the dissolved fraction, the particulate fraction is likely also important, and I feel like this would also help answer a similarly important question: are the nutrients in the forms they are because of the biological actors, or because of what the biological actors are sitting upon? This may also play a role in why some patches are in 'high abundance', and others are in 'low abundance', and thus would be collinear with cell abundance. Also, if a given sample was below detection, were they included in the analyses? They seem to be included in the figures, but would be good to know if they were also included in calculations, and if so which ones and how they were treated?

• **Response:** We agree with the reviewer about the potential importance of particulate nutrients in supraglacial environments, however, it was simply outside the scope of this study as we aimed toward understanding the dissolved phase. There is a large companion paper that investigates the mineralogy of the particles that comprise the surface impurities which is about to be submitted for review, and we will make reference to this following its submission and this second review of our manuscript. Investigation into the potential phosphorus input from particles in cryoconite holes has been investigated by Stibal et al. 2008. Text has been added in line 213 that states "Samples resulting below the LoD were considered 0 µM.".

RESULTS General comment: I think it would make more sense if the results section was more hypothesis-lead as well. Right now, it reads more like a list with some carefully chosen significant relationships scattered about and are difficult to understand how they relate to the overall picture.

• **Response:** The text in the results section has been completely reorganized with new sub-headings in order to make the reasoning and hypotheses clearer. Please refer to the results section in the updated manuscript (section 3) for the reworked text.

Line 176: In some ways, I feel like this opening sentence is really only validating the obvious. Transects were chosen based on the abundance of stuff covering them, and the first result is that more stuff was found in these patches covered with more stuff. I think that it would be more helpful to report it in this way such that it is setting up your experimental design rather than a unique result in its own right.

• **Response:** We believe that we need to make this is an important distinction. No quantification of the particulate content of the visible impurities was made. It is therefore important to state that not only did the algal cell abundance increase with the amount of visible impurities but that the differences in abundance were statistically significant. This also provides justification for the sampling method we employed.

Line 179-182: Why are correlations with DOC and DON reported here and not below? Why did you not compare with DOP? Also, while an interesting result, I feel like calling them 'highly significant' is a bit excessive, since the relationships (as far as I can tell anyway) seem to be based upon 3 comparisons apiece (averages of low, mid, high). Would Pearson correlations be the correct test here, or would it be better to test against the categories?

• **Response:** Since the reorganization of the results this correlation is now under the "Links between algal abundance and dissolved organic nutrients" subheading: please refer to section 3.2 of update manuscript. Please refer to above comment regarding the lack of DOP comparison. The Pearson correlation test was used as an initial test to illustrate a relationship between DOC/DON concentration and algal abundance. The term 'highly significant' has been removed.

Line 184 and elsewhere: Noting the number of samples that were over the LOD is great, but out of how many samples? What then happens to these below detection numbers. . .are they included in

calculations? Also, are some of these replicates or from the same patches? Are these also including cryoconites and supraglacial streams? The authors need to be more specific in their reporting of these data.

• **Response:** Text has been added at the start of results section 3.1 noting the total sample number for each nutrient for all five supraglacial environments sampled. Please see above comment with regards to values assigned to samples below the LoD. As stated earlier, sample locations were destructively sampled so the same patch was never sampled again. Two different samples each of the low, medium and high visible impurity ice were collected each sampling day, they are treated as individual samples in the data set.

Line 186: This is interesting. . .why do you think that NH4 was the dominant component of the DIN? Could this be from microbial ammonification of DON? I think this could be potentially also highlighted in a conceptual diagram!

• **Response:** Ammonification was not quantified, and so we are unable to definitively say the cause of the ammonium dominance. Telling et al. 2012 noted that the presence of NH4 in cryoconite hole samples might be an indication of active organic matter remineralization.

Line 194-200: Again, why do you not make comparisons with abundance and DOP? it seems central to what you are trying to find out, whether or not comparisons are 'significant' (in either case its interesting). It is also not clear which samples you are talking about . . .are they all pooled values for the ice sheet as a whole?

• **Response:** Please refer to above comment regarding the lack of DOP comparison. Section 3.1 of the results has been rewritten and text has been added explaining which samples are being referred to.

Line 196: 'The mean concentrations for the remaining 40 DIP concentrations [that were above the LOD] ranged from 0-0.7'... the lower limit should be 0.02, since that was the limit of detection, right?

• **Response:** Samples that fell below the LoD were considered to be 0  $\mu$ M, which is why our sample range begins at 0  $\mu$ M.

Line 198: 'DOP concentrations in cryoconite hole and supraglacial stream water fell below the LOD'. . .How do you mean this. . .that they fell below the LOD sometimes? In Figure 5, the average DOP for these two habitat types is around 7 uM. DON is a different story. . .Could it be that these two are being confused?

• **Response:** The text has been changed and the comment moved to line 262.

General comment: There are several mentions of nutrient ratios in the abstract and discussion. Why are these not discussed in the results? Also, where is figure 7?

• **Response:** Nutrient ratios have now been added to the result section 3.2 in response to the reviewer's comment. Figure 7 from the original version has been removed from the manuscript in response to reviewer 2 comments. Figure 7 is now the conceptual diagram.

DISCUSSION Lines 212-214: This information should be in the results, and it should be specified how they are calculated. For example, are these calculated for only surface ice environments? Furthermore, I think that the ratios of organic to inorganic nutrients would be potentially equally or more interesting to correlate with algal cell abundance than the absolute concentrations.

• **Response:** Percentages have been removed from the discussion and added to results section 3.1 with text added to explain which samples the percentages are referring to. See lines 230-231 in updated manuscript. The revised text discusses the increase of DON:DOP and DOC:DOP ratios with increasing visible impurities, lines 351-354 of discussion.

Line 215: Has this dominance been reported in other glacial systems?

• **Response:** Yes, dissolved organic dominance is commonly reported for cryoconite hole environments (Stibal et al., 2008; Telling et al., 2014). Lines 305-309 in updated manuscript describe dominance of dissolved organic nutrients in other glacial systems and its relation to microbial activity in the environment.

Line 222: Does Tedstone et al. 2017 actually report the timing of this shift in Nitrogen? Actually, has anyone reported this shift in nitrogen?

• **Response:** Text revised.

Lines 223-225: Similarly, how does this Williamson et al. (2018) paper support the shift in nitrogen phase? I think this needs to be rephrased/recast.

• **Response:** The authors have reworded the sentence. Please refer to lines 286-287 in updated manuscript.

Lines 226: But, these other impurities were not quantified, so it's difficult to say this for certain. For all we know, all the impurities could be ice algae! However, I think there may be some other papers showing this these days that you can cite. . .

• **Response:** Text changed in response to reviewer's comment. Yallop et al., 2012 has now been quoted as reporting a particle: cell ratio of 3:2 in the dark zone of the GrIS. Please refer to lines 292-293 in updated manuscript.

Line 227: There is a lot of talk of nutrients 'shifting' to the organic phase. But, it looks like to me that the concentration of DIN is basically the same for all the surface ice habitat types. Might the DON rather be accumulating through time from ice algae taking up DIN and subsequently 'leaking' DON into their habitat, rather than the DIN pool shifting? It would really be nice to see these relationships over time.

• **Response:** The authors would like to clarify that the use of the term 'shift'. With regards to the nitrogen nutrient pool the author's use of 'shift' has to do with the snow and ice core data that show a dominance of DIN with little to undetectable levels of DON. Yet, as the season progresses the dissolved nutrient pool is dominated by the dissolved organic phase, showing that something is occurring at the ice surface to cause the nitrogen pool to change. The authors agree that it is very likely the ice algae up taking the DIN, utilizing it and producing DON, which is the main argument of this paper: ice algae are the drivers in this conversion of nutrients.

Line 228: Furthermore, the big differences in organic/inorganic nutrients with algal biomass seems to only apply to nitrogen, and I think it is important that this distinction is made. Why would this not apply to phosphorus? This should be discussed in detail, and the authors should be more specific whether they are talking about 'nitrogen' or indeed 'nutrients' (ie nitrogen + phosphorus) elsewhere in the manuscript.

• **Response:** The revised text hopefully makes this clear.

Line 230: Do the data really suggest 'efficient' conversion? I think at best there is a correlation between cell counts and organic nutrients, but no data that points directly to conversion, and definitely no data that would suggest that the process is efficient (for example, the DIN concentration seems unchanged with increasing cell abundance). Furthermore, why do you think the same would not be seen for DOP?

• **Response:** Text changed in response to reviewer's comment and the term conversion has been removed. Please refer to line 268 in updated manuscript. Discussion of changes in DOC:DON:DOP ratios can be found in Section 4.4.

Line 232-233: I think this information belongs in the results section. Furthermore, Figure 7 is mentioned for the first time here. Maybe would it be better to put this in supplementary information if it is not going to be used to support the main results? Individual data points could also be superimposed onto bar figures (e.g. 'jittered' points in ggplot2) to illustrate variability between categories, if that is the goal.

• **Response:** Figure 7 from the original version and linear regression relationships have been removed from the manuscript in response to reviewer 2 comments.

Line 239: 'Demonstrate' is strong in this case. . . perhaps 'suggests'?

• **Response:** This sentence has been deleted in revised text.

Line 240-241: Are ice algae assemblages the main producers of dissolved organic nutrients stocks in freshwater and marine ecosystems? Recast this text.

• **Response:** This sentence has been deleted in revised text.

Line 242: Do the ice algae really 'rapidly' take up inorganic nutrients? If there are some numbers to back this statement that is great, but I think this cannot be said without some support.

• **Response:** This sentence has been deleted in revised text.

Line 243: I still think that it would help to somehow organize these sources in a diagram to help guide your thinking and the readers comprehension. What forms of inorganic nitrogen is deposited on the ice sheet and how? How about organic forms? Phosphorus?

• **Response:** The authors have produced a conceptual diagram in response to reviewer's comment. The diagram depicts likely nutrient inputs to supraglacial environments, ice algae producing dissolved organic N, P and C and inefficient remineralization by heterotrophs. The diagram is simple due to the fact that many aspects of nutrient input, cycling and export in the Dark Zone of the GrIS still remain unknown and was one of the main objectives of this paper, to produce a preliminary dataset of dissolved inorganic and organic nutrients for this region. The authors fear

that by making this diagram overly detailed it could be misleading as not enough research has been done in the Dark Zone. Please see Figure 7 for conceptual diagram.

Line 245: Can also be breakage, leakage, or lysis, for example. . .what about extracellular processes?

• **Response:** The authors included 'decomposition of the ice algae' to account for the breakage, leakage or lysis input of dissolved organic nutrients. Extracellular processes such as the production of EPS is addressed in section 4.3 of the discussion.

Line 248: Does bacterial carbon production equate to nutrient-transformation processes like ammonification? If bacteria are really that sparse, I think you could alternatively think that they are really efficient, since they seem to be producing measurable ammonium in excess of uptake.

• **Response:** It is possible that depletion of nitrate and higher levels of ammonium could suggest ammonification, but it would only be speculation within the constraints of this manuscript. Furthermore, the authors would also like to clarify that the manuscript comments on bacterial production rates in comparison to net primary production, not bacterial abundance. Nicholes et al. (2019) reports bacterial abundance as  $3.3 \pm 0.3 \times 10^5$  for surface ice samples taken during the same field campaign as the present study. This shows that bacteria are abundant, but not active.

Line 251-254: 'Reduced capacity' is interesting wording. . .were they at higher capacity at some point? I think the production of ON is just outpacing mineralization

• **Response:** This sentence has been deleted in revised text.

Lines 257-259: This is interesting that all of these different habitat types studied by Stibal et al. (2008) also had the organic forms dominate. Why do you think this was not the case for Nitrogen in the supraglacial streams and cryoconites from this study, while it also it seems to hold true for phosphorus?

• **Response:** The authors believe that there could be differences between the two studies due to retention by surface ice microbial communities. One conclusion of this manuscript is a retention ability by the microbes in the surface ice to hold dissolved organic nutrients at the surface via the production of EPS. As EPS contains N, it is likely that N is being retained at the ice surface as opposed to being transported through the water table. DOP is also exuded in the form of EPS, but actually the difference in DOP and DIP in supraglacial streams is not statistically different. The concentrations of DIN/DON and DIP/DOP in cryoconite hole water are not statistically different either.

Line 271: Are ice algae producing EPS? Has anyone tried to quantify this?

• **Response:** To the authors knowledge, quantification of ice algal production of EPS has not been conducted, but Yallop et al. 2012 identified EPS in surface ice samples dominated by ice algae.

Line 279-280: is it possible that DON and DOP are also 'over-wintering' on top of the icesheet? Could any of this be 'leftovers' from a previous season?

• **Response:** We believe that some DON and DOP can remain in the ice surface at the end of the ablation season and remain frozen until the next season. We use Musilova et al. 2017 to provide

an example that this has already been shown for DOC. Please refer to lines 332-340 of updated manuscript.

Line 280: This sentence is vague. . .what exactly about the export of dark zone DOM is unknown. . .the character. . .the quantity?

• **Response:** This sentence has been deleted in revised text.

Line 285: The Redfield Ratio was certainly generated using data from marine systems, but I think its utility over the last decades has been in providing a point of comparison. However, I think it also deserves clarification that the Redfield Ratio is the average molar ratio of biomass under balanced growth. Do we know the elemental composition of ice algae under balanced growth, and how it compares to Redfield Ratio? I'm also not sure that I understand the purpose of the text that follows. While there is certainly a lot of variability across aquatic habitats in dissolved N:P ratios from cold regions around the world (and elsewhere), I'm not sure how useful it is to bring up these numbers here. Furthermore, it is not clear if the ratios from the cited studies are also using the organic fractions-only as done in this study (my guess is that this is not the case). If the purpose of this text was to (presumably) link the reported N:P ratios discussed in the paragraph below to the literature, this makes comparisons difficult, and calls into question the need for this text, or at least would suggest that it needs to be revised to fit the authors' purpose.

• **Response:** We agree that the text following the Redfield Ratio in the original manuscript may cause confusion for the readers and it has been removed. We know of no published literature on the elemental composition of ice algal under balanced growth.

Line 295: This is the first time DOC:DON:DOP ratios have been reported besides in the abstract. . ..I did not see it in the introduction, methods, or results that you planned to look at these ratios.

• **Response:** Text has been added to the updated manuscript to include ratios in the results section 3.2. Line 130 of the introduction now describe our intent to investigate nutrient ratios in the manuscript. We do not believe that it is appropriate to include the ratios in the methods section.

Lines 298-300: Why are you making nutrient ratios for the organic form of these nutrients? Wouldn't you expect that algae would be taking up the inorganic forms primarily (especially NH4)? I feel like these ratios might not be accurately approximating availability for algae, and thus I'm not sure that, based on comparing these ratios with the Redfield Ratio alone, that we can say that the system is P-limited. I think it needs to be carefully explained in the text why this would be the case.

• **Response:** Please see revised Discussion Section 4.4.

Lines 301-304: Would it be possible to more rigorously investigate this statement of different slopes of CP and CN over algal abundance? I think that this could be interesting if better developed, but as written it seems more of an afterthought.

• **Response:** Please see revised Discussion Section 4.4.

Line 313: Is cryoconite the same as the particles talked about in the introduction?

• **Response:** Cryoconite is part of the LAIs discussed in the introduction. Cryoconite holes can melt out or be flushed out throughout the season which causes the cryoconite debris to be washed over the surface. The particles referred to in the introduction describe dust ablating out from meteoric ice as reported by Wientjes et al. 2011.

Lines 326: In order to be able to say 'rapid uptake of dissolved nutrients', you need to have data on the uptake rates to compare. You also do not report rates of organic production.

• **Response:** "Rapid" has since been removed from the sentence. We use the term "high production of dissolved organic production" to refer to the high concentrations of C, N and P produced not to imply any rate at which the production is occurring.

Lines 328-329: These production data are also assumed to hold true here, as production wasn't investigated in this work. Also, why would it be inefficient. . .because there are leftover organic nutrients?

• **Response:** An efficient microbial loop has similar rates of NEP and secondary production, which results in more balanced concentrations of dissolved organic and inorganic nutrients. The fact that there is such a dominance of dissolved organic nutrients implies that remineralization rates are low/inefficient. Nicholes et al. (2019) is cited here because they determined a 30:1 ratio for the same surface ice samples reported in this study, as this manuscript focuses on the geochemistry.

Line 332: Was this the case for phosphorus? Also, I think that the notion of this retention being due to EPS is too speculative to say it this way.

• **Response:** We believe that the retention of dissolved organic nutrients via the production of EPS is a viable hypothesis. Please see revised Discussion Section 4.4.

Line 334: This is vague and repeated from line 280.

• **Response:** This sentence has been deleted in revised text.

**TECHNICAL CORRECTIONS**

Line 23: Comma after 'nitrogen' not necessary

• **Response:** Text changed in response to reviewer's comment.

Line 30: Should there be spaces between values and "Gt"?

• **Response:** Spaces have been added between values and units throughout the updated manuscript.

Line 36: Similarly, should there be a space between "30" and "km"? This should be fixed throughout.

• **Response:** Please refer to above comment.

Line 160: HCl

• **Response:** Text changed in response to reviewer's comment.

Line 214: comma after "To date"

• **Response:** Text changed in response to reviewer's comment.

Line 241: here and elsewhere, references should be ordered.

• **Response:** Text changed in response to reviewer's comment.

**Author's Responses to Reviewer 2:**

**Overall response:** we would like to thank the reviewer for the helpful and constructive review. We have made extensive changes to the text, particularly the Discussion, in line with the commentary below and that of the other Reviewer. We feel that the manuscript has been significantly improved as a consequence.

**Reviewer comments and responses.**

This paper provides novel information on the chemistry of supraglacial ecosystems. The main finding is that most of the dissolved N and P in these environments is in organic rather than inorganic forms. The authors use their chemical data in concert with measurements of algal cell abundance to make inferences about the role of microbes in supraglacial nutrient cycling. The paper is generally well written and would be of interest to biogeochemists, and to a lesser extent, hydrologists and glaciologists, working in ice-covered ecosystems. There are several sections of the paper that I felt were overly speculative, especially with regards to rates and mechanisms of nutrient retention. In addition, I believe that the authors could better reconcile their findings with previous literature on OM production in supraglacial environments. As a result, I think the paper needs some important revisions before it should be considered for publication in Biogeosciences Discussions. I have provided comments and editorial suggestions below that I hope will be helpful for revising the paper.

• **Response**: The authors would like the thank the reviewer for their in-depth assessment of our manuscript and for providing beneficial comments for the restructuring of the manuscript. We direct them to our responses to each individual question below.

Line 99: It would be appropriate to report the number of samples collected for each habitat type somewhere in this section.

• **Response**: We now include the sample sizes of each habitat.

Line 101: How were sample locations classified into low, medium, and high impurity categories? The figure gives a sense of the density of impurities but there is no indication of whether there was some quantitative aspect to the process (i.e. number of impurities per unit area) or whether the process was

wholly subjective. Also, the nature of the impurities is not well described – are they mineral, biological, or a mixture of both (such as the material found in cryoconite holes)?

• **Response:** Sample locations were determined visually as the difference in the impurity loadings was quite apparent, as the authors tried to show in Figure 2. There was no quantitative process conducted on the ice surface prior to choosing the sample location, however Figure 3 reinforces the validity behind choosing sights visually as there was a significant increase in algal abundance between the low, medium and high visual impurity ice. There was no further analysis of the impurities beyond Yallop et al., 2012, who quantified a 3:2 particle: cell ratio for their samples collected in the Dark Zone. Furthermore, a companion paper is being produced that investigates the mineralogy of the impurities collected.

Lines 179-181: The comparisons between algal cell abundance and organic nutrients are inconsistent. Algal cells and DOC are compared by regression, algal cells and DOC and DON are compared by pearson correlation, and algal cells are not compared at all to DOP. Moreover, these tests do not provide any information about the differences in the relationship between different habitats.

• **Response**: Only Pearson correlation is reported in the revised manuscript, and only significant relationships were reported. DOP did not correlate significantly with algal abundance. ANOVA analysis is included to provide information about the differences in nutrient concentrations between habitats.

Line 184: What was the LoD for DON? Are the sample numbers you report (54 DON samples, 41 DIN samples) out of the 70 samples you included in the data for Figure 4? Also, what value did you use for all of the samples that were below the LoD – half of the LoD or some other value?

• **Response**: LoD for DON is 0.87 µM and is included in the revised text. Lines 229-230 have been added for clarification about the number of samples for each test. Values below the LoD were considered to be 0 µM, line 214 has been added for clarification.

Line 194: What was the LoD for DOP? There were 74 DOP samples above the LoD, however in the legend for Figure 5 it appears that only 70 DOP samples were included in the figure.

• **Response**: LoD for DOP is 0.02  $\mu$ M and is included in the revised text, as are the correct number of samples.

Line 230: How do you get information about conversion rates from the concentrations you measured?

• **Response:** This sentence has been deleted in revised text.

Lines 232-234: This regression plot is not an effective way to analyze the relationship (or lack thereof) between DOC and algal abundance. The fact that there is any positive relationship is based on the single outlier in the upper right hand corner of the graph. If you removed that outlier, it appears that there would be a negative relationship between DOC and algal cell abundance (or, at best, no relationship). If there is, in fact, no relationship between algal cell abundance and DOC, that does not seem to support your statement that you "interpret these data to demonstrate that ice algal assemblages are the main producers of dissolved organic nutrient stocks within the melts surface ice. ..." (line 239). This may well be true but

it is not what these data show. There are other possible explanations for the lack of relationship between DOC and algal cell counts including that you are comparing data collected across a full month and the relationship may change over the melt season.

• **Response:** We would like to thank the reviewer for pointing out this oversight on our part. The plot and linear regression analysis have been removed.

Lines 234-237: Similar to the comment above, this explanation for the lack of a relationship between algal cell abundance and DOC would be more convincing if it detailed more specifically how these variables could become decoupled rather than just invoking the "highly dynamic nature of the environment" where solutes and gases move around.

• **Response**: We agree and have further elaborated on weathering crust dynamics have been included.

Lines 267-274: It is surprising that cryoconite holes have low stocks of dissolved organic nutrients compared to surface ice. Past research has focused on cryoconite holes as hotspots of C fixation in autotrophic supraglacial environments (e.g. Anesio et al., 2009, Global Change Biology). If this were the case, it seems that the abundant production in cryoconite holes would be reflected in dissolved organic nutrient concentrations, but that is not what these data show. Does this suggest that surface ice habitats are potentially more important for autotrophic production or is there another explanation? Also, if you invoke EPS, which is known to occur in cryoconite holes, as the mechanism by which nutrients are retained in surface ice, wouldn't this also be true for cryoconite holes and drive up dissolved organic nutrient concentrations in the same way in those habitats?

• **Response:** We thank the reviewer for this commentary. We have made substantial revision to the Discussion and decided to concentrate mostly on differences between macronutrient concentrations in the melting surface ice environments. We felt that a discussion of processes in cryoconite holes detracted from the main message of the paper, and so have not included these types of ideas here. However, we fully agree with the reviewer that this is a very interesting idea. It is likely that the melting surface ice does fix more carbon by dint of the greater surface area, but this is not the main thrust of this paper.

Line 277: I don't find the argument for a "large pulse of dissolved organic nutrients" particularly convincing. Particulate organic nutrients are hardly mentioned in this paper. It seems like a more parsimonious explanation for the loss of organic nutrients produced in supraglacial habitats is that they are exported downstream, at least partially, in particulate forms.

• **Response:** We agree and have downplayed this idea in the revised manuscript.

**Editorial Suggestions**

Line 77: change "to accumulate" to "accumulation"

• **Response:** Text changed in response to reviewer's comment.

Line 101: add "of" after "amounts"

• **Response:** Text changed in response to reviewer's comment.

Line 140: It would be helpful to define the acronym TON. I presume that it represents total oxidized nitrogen here but this acronym is commonly used to refer to total organic nitrogen (dissolved + particulate ON) so you should be clear about how it is being used.

• **Response:** Text changed in response to reviewer's comment, line 186 now reads: "NO2- and total oxidized nitrogen (TON) (NO2- + NO3-)...".

Lines 179-180: This sentence refers to data shown in Fig. 7 (currently referenced on line 233), which should be renumbered to Fig. 4 and cited here.

• **Response:** Figure 7 has been removed from the manuscript after review from the above comment and has been replaced with a conceptual diagram.

Lines 189: "increase" should be "increased" to be consistent with the rest of the results which are in the past tense.

• **Response:** Text changed in response to reviewer's comment.

Line 234: Suggest changing "counts were" to "abundance was" since DOC is not plural.

• **Response:** Text changed in response to reviewer's comment.

Dissolved organic nutrients dominate in Nutrient cycling in melting surface ice of the Dark Zone (Greenland Ice Sheet) supraglacial environments of the Dark Zone of the Greenland Ice Sheet

1 Alexandra T. Holland1, Christopher J. Williamson1,2, Fotis Sgouridis4, Andrew J.

2 Tedstone1, Jenine McCutcheon5, Joseph M. Cook6, Ewa Poniecka7, Marian L. Yallop2,

3 Martyn Tranter1, Alexandre M. Anesio1,3, and The Black & Bloom Group\*

4 1Bristol Glaciology Centre, School of Geographical Sciences, University of Bristol, BS8 1HB, UK.

2School of Biological Sciences, University of Bristol, 24 Tyndall Avenue, Bristol, BS8 1TQ, UK.

6 3Department of Environmental Science, Aarhus University, Roskilde, 4000, Denmark.

4School of Geographical Sciences, University of Bristol, Bristol, BS8 1RL, UK.

8 5School of Earth and Environment, University of Leeds, Leeds, LS2 9JT, UK.

[revised manuscript text omitted]

64 Both snow and bare ice albedo are reduced by light absorbing impurities (LAIs), of, which includeboth biological 65 and mineralogical originsubstances (Gardner and Sharp, 2010), which . Types of LAI includee atmospheric dust 66 and black carbon, cryoconite, and particulates within the meteoric ice that melt out during the ablation season 67 (Warren and Wiscombe, 1980; Warren, 1984; Warren and Wiscombe, 1985; Gardner and Sharp, 2010; Wientjes et al., 68 2012;Cook et al., 2016a).—. The importance of biological LAI, specificallyparticularly Streptophyte ice glacier algae, 69 that which form significant algal blooms in surface ice environments during summer ablation seasons, as a factor in 70 71 "bioalbedo", which is derived from the original term "biological albedo reduction" (Kohshima et al., 1993;Cook et 72 73 during the bloom (up to  $\sim 10^4$  cells ml-1 surface ice) and the heavily pigmented nature of the ice algal cells, which 74 includeing production of a unique dark UV-VIS absorbing pigment, UV-VIS absorbing-purpurogallin-type pigment, 75 that, purpurogallin, in the ice algae, which is postulated to provides photo-protection from the extreme solar 76 radiation in supraglacial environments, and the abundance of cells apparent achieved during bloom progression (up 77 to ~104 cells ml+ surface ice) (Remias et al., 2012; Williamson et al., 2018).—\_Tedstone et al., (2017) concluded that 78 ice algal blooms are the main factor responsible for inter-annual variability in the extent, magnitude and duration of 79 the Dark Zone, which -and seem to be regulated by climatic drivers, including the June-July-August sensible heat 80 flux anomaly and the timing of snow-line retreat.—. The spatial extent of heavy ice algael blooms may also be linked 81 also to the availability of mineralogic LAIs, such as late Holocene dust particles melting out of the ancient meteoric 82 83 (Tedstone et al., 2017). ... Furthermore, within the Dark Zone, Vallop et al., (2012) noted significant spatial 84 heterogeneity in the ice algal surface ice colonisation, varying on length scales of cm to tens of meters. 85 Carbon, nitrogenN and phosphorusP are essential for all living organisms, as they provide ing the basis for cellular 86 mass and all metabolic activity (Redfield et al., 1963; Hessen et al., 2013).-. As eCarbon is usually in ready supply in 87 surface ice environments, both from the atmosphere and from bubbles trapped in snow and ice, and so nitrogen and 88 phosphorus are more likely the limiting factors for growth and activity of microorganisms (Stibal et al., 2009;Lutz et 89 al., 2017). - Carbon is readily available in these environments for two main reasons. First, as snow forms in the 90 atmosphere it seavenges nutrients in the form of trace gasses and incorporates them, in the dissolved inorganic 91 phase, into the snow crystal (Kuhn, 2001). The snow accumulates on the ice sheet surface, and during the ablation 92 season, melts and releases dissolved inorganic carbon, to the supraglacial environments (Fig. 7). Second, as the ice 93 surface is constantly open to the atmosphere during the main ablation season, gas exchange can occur across the air-94 water interface (Liss, 1973). Carbon, in the form of CO2, dissolves in water pooled on the ice surface and becomes 95 bioavailable to microbes in the form of bicarbonate (HCO3-), carbonate (CO3-) and CO2 (Liss, 1973). As carbon is

96 usually in ready supply in surface ice environments, \_nitrogen and phosphorus are more likely the limiting factors 97 for growth and activity of microorganisms (Stibal et al., 2009;Lutz et al., 2017).Like carbon, Bioavailable forms of 98 N are less readily available, being largely confined to  $NO_3$  and  $NH_4^+$  in dry and wet deposition from the atmosphere 99 (Wolff, 2013), and from snow- and ice-melt (Telling et al., 2011). Dissolved inorganicnitrogen is seavenged from 100 the atmosphere by snowfall and released to supraglacial environments by snowpack melt as its main input source 101 (Fig. 7) (Kuhn, 2001). Yet, even though N2-comprises a large portion of the atmosphere, it is not easily bioavailable 102 and not all photosynthetic organisms are capable of fixing it from the air (Falkowski and Raven, 1997). Telling et 103 al., (2012) even reported that the importance of nitrogen fixation for microbial growth decreased with distance from 104 the margin on the GrISTelling et al., (2012). Therefore, gas exchange over the air water interface, that assists 105 earbon deposition, is not equally beneficial for nitrogen. Dissolved inorganic phosphorus phosphorous (DIP) is 106 typically the least available nutrient in supraglacial environments (Stibal et al., 2009; Stibal et al., 2008b), assince it 107 is a largely rock--derived and -mineral and-is only released by chemical and physical-weathering or bio-mining 108 (Stibal et al., 2009; Stibal et al., 2008b) of rocks. P sources Consequently, in remote glaciated environmentsareas, 109 such as the Dark Zone, phosphorus input is limited are largely- confined to the small quantities of particulates 110 deposited from the atmosphere and the melt out of debris in snow and ice (Wientjes and Oerlemans, 2010). 111 -The presence of such large-scale algal blooms in the Dark Zone, with cell abundances as high as 8.5 x 104 cells ml- 112 113 However, the the current literature, which suggests that supraglacial environments in the Dark Zone, similar to those 114 found in Svalbard, the margins of the Greenland Ice Sheet and Antarctica, are extremely oligotrophic (Stibal et al., 115 2008b;Stibal et al., 2009;Telling et al., 2011;Telling et al., 2012;Hawkings et al., 2016;Wadham et al., 116 2016;Bagshaw et al., 2013) .-... Mean A comprehensive review of dissolved inorganic -nitrogen (DIN) concentrations 117 in Greenland ice are was conducted by Wolff (2013), who reported that mean dissolved inorganic nitrogen 118 eoncentrations in ice cores of arc-1.4 µMµmol++, with NO3- and NH4+ nitrate and ammonium composing 0.97 µM 119 µmol]+-and 0.3945 µMµmol]+, respectively (Wolff, 2013).—. There are relatively few measurements of nutrient 120 concentrations in the surface ice environments inof the Dark Zone (Telling et al., 2012; Wadham et al., 2016), but the 121 a. Values of Aaverage NO3-nitrate concentrations in surface ice nearalong the K Transect east of Kangerlussuaq, 122 which passes through the Dark Zone, has been are were reported to be  $0.6 \pm 0.1 \mu M \mu mol l^+$  for surface ice located 123 between 17-79 km from the ice sheet margin (Telling et al., 2012), whilest. DIp PhosphateP concentrations are were 124 reported as being below the detection limit, 0.33µM P (Telling et al., 2012)..... In contrast, dissolved inorganic 125 nitrogenDIN -concentrations in snow sampled before the start of the ablation season at the margin of the GrIS had 126 higher were reported as higher than surface ice-concentrations, with an average of 1.4 µMmol 1-1 (Telling et al., 127 2012), similar to those of Wolff (2013). Hence, there is no real evidence that neither N nor P concentrations in snow 128 and ice sampled in the vicinity of the Dark Zone are higher than for average Greenland ice. - We anticipate that this 129 average snow concentration may be an upper limit for the Dark Zone during the height early of the ablation season, 130 given the high concentrations of ice algae that occur during blooms.

[revised manuscript text omitted]

scale, yet no studies to date have examined variability on these these length scales. We contend that it is important to

- 168 determine the concentrations and relative proportions of dissolved inorganic and organic nutrients in melting surface
- 169 ice environments of Dark Zone, particularly during Streptophyte iceglacier algae blooms, since a knowledge of both
- 170 DIN, DON, DIP and DOP may be crucial to better understand how glacier ice algae and bacteria can retain, utilize
- 171 and recycle their limited nutrients to sustain the large-scale blooms observed in this region of the Greenland Ice
- 172 Sheet. -. Yet, dissolved macronutrients tend to concentrate in the dissolved organic phase (Stibal et al.,
- 173 2008b;Telling et al., 2014), suggesting an imbalance in the uptake and remineralization of dissolved inorganic
- 174 nutrients in cryoconite hole environments and as a consequence, dissolved macronutrients tend to concentrate into
- 175 the dissolved organic phase (Stibal et al., 2008b;Telling et al., 2014). To date, dissolved organic nitrogen
- 176 concentrations in the Dark Zone of the GrIS have only been reported in two studies (Telling et al., 2012; Wadham et
- 177 al., 2016), yet neither focus on ice populated by Streptophyte ice algae. Furthermore, and phosphorus concentrations
- 178 for surface ice environments in the Dark Zone have not been reported to date., and Wwe contend that this may be an
- 179 important omission in our understanding of Dark Zone microbial nutrient cycling, specifically as it relates to the
- 180 extensive Streptophyte ice algae blooms. Knowledge of both the dissolved inorganic and organic phases of
- 181 nitrogen, phosphorus and carbon may be crucial to better understand ice surface nutrient cycles and how ice algae
- 182 and bacteria can retain, and recycle utilize and recycle their limited nutrients to sustain the large scale blooms
- 183 observed in this region of the Greenland Ice Sheet.
- 184 The aims and objectives of this study, therefore, are threefold—\_First, we aim to quantify dissolved nutrient
- 185 concentrations in the supraglacial environments of the Dark Zone during the peak ablation season—.Second, we
- 186 determine the relative importance abundance of dissolved inorganic and organic nutrients during the peak ablation
- 187 season when microbial recycling is likely to have the greatest influence on the dissolved inorganic and organic
- 188 ratios....LastFinally, we investigate if there are systematic changes in the relative proportions of dissolved
- 189 macronutrients during differences in nutrient concentrations in highly increased colonizationed of melting surface
- 190 ice, which might shed light on the limiting nutrient on algal bloomsenvironments compared to others with lower
- 191 levels of ice algal biomass.
- 192

**193 **2.** Methods**

**194 2.1 Field Site and Sampling**

[revised manuscript text omitted]

**238 2.2 Analytical Methods**

- Algal cell abundance was quantified using a Fuchs-Rosenthal haemocytometer (Lancing, UK) on a Leica DM 2000
- 240 epifluorescence microscope with attached MC120 HD microscope camera (Leica, Germany)—\_For samples
- containing sufficient cell abundance, a minimum of 300 cells were counted to ensure adequate assessment of
- assemblage diversity (Williamson et al., 2018).
- 243 TDN (total dissolved nitrogen) is the sum of DIN (dissolved inorganic nitrogen) and DON (dissolved organic
- 244 nitrogen). DIN species include NH4+, NO2- and NO3- and were quantified as follows.- First, NH4+ was quantified
- 245 spectrophotometrically using a Lachat QuickChem® 8500 Series 2 Flow Injector Analyzer (FIA; QuickChem®

- 248 determined by dividing the standard deviation of the response of the calibration curve by the slope of the calibration
- 250 considered 0  $\mu$ M for all analyses. Precision was ±2.1%, and accuracy was +8.5%, as determined from comparison
- with a gravimetrically diluted 1000 mg  $L^{-1}$  NH4+-N certified stock standards to a concentration of 1.1  $\mu$ M. (Sigma
- 252 TraceCERT®).—. Second, NO2- and total oxidised nitrogen (TON) (NO2- + NO3-) were quantified
- 253 spectrophotometrically using a Gallery Plus Automated Photometric Analyzer (Thermo Fisher Scientific, UK)-
- This combination of analysis allows the original  $NO_3^-$  concentration to be determined by subtracting  $NO_2^-$  from TON-.
- 256 TDN-(total dissolved nitrogen) is the sum of DIN and DON, and TDN was determined after by digesting the samples
- with a potassium persulfate, sodium hydroxide and boric acid reagent and autoclaving at 121°C for 30 minutes and
- 258 measuring as TON as above (Grasshoff et al., 1999). This process causes the oxidation of organic nitrogen
- 259 compounds, which can then be measured as TON as above. Purification of the potassium persulfate was conducted
- 260 via recrystallisation in order to remove any N contamination. DON was then estimated by the difference of DIN
- 261 from TDN (DON= TDN DIN). the original TON and NH4+ from the TDN of the persulfate digestion (DON=TDN-
- 262 NH4+-NO2--NO3-). Measurements were based on the hydrazine-sulfanilamide reaction method measured at
- 263 540nm...DON was then estimated by subtracting DIN from TDN (i.e. DON= TDN-DIN).... The LoD wasere 0.14
- 264  $\mu$ M (NO2-), 0.64  $\mu$ M (TON) and 0.87  $\mu$ M (TDN/DON)—Precision was ±0.87% (NO2-), ±1.17% (NO3-) and ±0.63%
- 265 (TDN/DON), and accuracy was -4.04% (NO2--), -8.07% (NO3-) and -5.7% (TDN/DON), as determined from
- 266 comparison with gravimetrically diluted 1000 mg L-1NO2-N and NO3-N certified stock standards to a
- 267 concentration of 0.71  $\mu$ M (NO2-), 1.4  $\mu$ M (NO3-) and 7.1  $\mu$ M (TDN/DON) (Sigma TraceCERT®).
- TDP (total dissolved phosphorus) is the sum of DIP (principally  $PO_4^{3-}$ ) (dissolved inorganic phosphorus, principally
- 269 PO43-) and DOP (dissolved organic phosphorus). \_\_\_ The same persulfate digestion method described for TDN was
- 270 used to measure TDP as PO43-...DOP is determined by the subtraction of DIP in the undigested sample from the
- 271 TDP in the digested sample. PO43- in both the-undigested and the digested samples was quantified using a Lachat
- 272 QuickChem® 8500 Series 2 Flow Injector Analyzer (FIA; QuickChem® Method 31-115-01-1-I) using the

- 274 sample from the TDP in the digested sample (i.e. DOP = TDP-DIP)—. The LoD was 0.02  $\mu$ M (PO43- and
- 275 TDP/DOP)....Precision was  $\pm 1.6\%$  (PO43-) and  $\pm 3.1\%$  (TDP/DOP), and accuracy was  $\pm 2.3\%$  (PO43-) and  $\pm 5.0\%$
- 276 (TDP/DOP), as determined from comparison with gravimetrically diluted 1000 mg  $L^{-1}$  PO4-P certified stock
- 277 standards to a concentration of 0.65  $\mu$ M (Sigma TraceCERT®)—.
- All DIN, DON, DIP and DOP data were water blank-corrected using values from the respective field procedural
   blanks (Table 1).
- 280 DOC concentrations were quantified using a Shimadzu TOC-L Organic Carbon Analyzer, with a high sensitivity
- catalytic combustion (680°C) of dissolved organic carbon to carbon dioxide, which was then measured by infrared
- absorption—. The LoD was 9.5  $\mu$ M—. Precision was  $\pm 2.4\%$  and accuracy was -5.9%, as determined from
- $^{284}$  comparison with gravimetrically diluted 1000 mg L-1 TOC certified stock standards to a concentration of 83.3  $\mu$ M
- 285 (Sigma TraceCERT $\mathbb{R}$ ).

**286 2.3 Data Analysis**

**287**

- All measurements below the LoD were considered to be 0 for all statistical analyses. All DIN, DON, DIP, and DOP 288 289 and DOC data were water blank-corrected using values from the respective field procedural blanks (Table 1)---. 290 Additionally, all blank corrected values that were negative were assumed to be 0 for all statistical analyses. SAH 291 statistical analysis wwwas performed in RStudio v.1.1.414 (RStudio, Inc 2018)-\_\_Identification of statistical 292 differences between the nutrient content, DOC concentrations and algal cell abundance between din different 293 habitats was achieved using 1-way analysis of variance (ANOVA) or t-test comparisons, with post-hoc Tukey HSD 294 analysis applied to all significant ANOVA results-.. Linear regression models and Pearson's product-moment 295 correlations were used to identify correlations between DON, DOC and algal cell abundance. -Homogeneity of 296 variance and normality of distribution were tested prior to all parametric analyses, and model assumptions were 297 verified by examination of model criticism plots-
- 298

**299 3. Results**

- 301 3.1 Dissolved nutrient concentrations in surface ice with differing levels of visible impuritiesAlgal Cell
   302 Abundance
- Supraglacial environments are extremely oligotrophic, making the measurements of dissolved nutrients difficult.....
   Dissolved nutrient concentrations reported in previous studies of supraglacial environments are typically atbelow or

1
 2
 308 Dissolved organic concentrations were significantly higher than dissolved inorganic concentrations for nitrogen and

**309** phosphorus.—. AboutSome 93% of the total dissolved nitrogenTDN was in the form of DON and about 67% of the

**311** concentrations for the three surface ice habitats range from  $5.10-14.0 \mu$ M, while those for DIN range from 0.62-1.0

314 Jum (Fig. 4, Table 1). While Similarly, -DOP concentrations were usually at least twice those of o times higher than

B15 DIP concentrations for the three ice surface samples habitats, with mean mean-values ranging from 0.10-0.1515  $\mu$ M

and 0.03-0.0707 µM respectively (Fig. 4, Table 1).... T-tests revealed significant differences between DON and DIN

317 in all three surface ice habitats five supraglacial environments except cryoconite hole water (low:  $t_{36}$ =3.6, p<0.001,

**318** medium:  $t_{36}$ =5.3, p<0.0001, high:  $t_{36}$ =7.4, p<0.0001, stream:  $t_{36}$ =-2.6, p<0.01) (Fig. 34) and DOP concentrations as

319 significantly higher than DIP concentrations for all three surface ice habitats (low:  $t_{36}=3.1$ , p<0.01, medium:  $t_{36}=2.1$ ,

**320** p < 0.05, high:  $t_{36} = 3.7$ , p < 0.001) (Fig. 45)... DONC and DOCN concentrations in the three surface ice habitats

321 showed clear trends with increasing visible impurities (Fig. 34 & 56)..... DON concentrations increased significantly

**322** from low to medium and low to high visible impurity loadings ( $F_{4,71}$ =19.8, p<0.05,  $F_{4,71}$ =19.8, p<0.001,

323 respectively), while DOC concentrations increased significantly in ice with high and low visible impurity loading

324 ( $\underline{F_{4.74}}=6.8, p<0.01$ ). Algal cell abundance increased significantly with the amount of visible impurities seen on the

325 ice surface, as shown in Figure 3 ( $F_{2,54}$ =26.1, p<0.0001). The mean (± standard error) concentrations in the three

326 surface ice habitats were:  $99.5 \pm 23.9$  cells mL-1 for ice with low visible impurities,  $3850 \pm 530$  cells mL-1 for ice

327 with medium visible impurities and  $9800 \pm 1570$  cells mL-1 for ice with a high loading of visible impurities.

328 Significant Pearson's product-moment correlations were apparent between average algal cell counts and DON and

329 DOC surface ice concentrations ( $t_3$ =3.5, p<0.05, r=0.9 and  $t_3$ =5.4, p<0.01, r=0.95, respectively). A significant

330 linear relationship was apparent between algal cell counts and DOC in surface ice habitats ( $R^2=0.1$ , p<0.01, n=57).

331 Highly significant Pearson's product moment correlations were apparent between average algal cell counts and

**332** DON and DOC surface ice concentrations ( $t_3=3.5$ , p<0.05, r=0.9 and  $t_3=5.4$ , p<0.01, r=0.95,

333 respectively).Comparison of DOP surface ice concentrations and algal cell counts were not significant.

334

**335 3.2 Links between algal abundance and dissolved organic nutrients Nitrogen**

336 ANo quantification into the mineralogic composition of the visible impurities was conducted., but algal cell

abundance, which ranged from 90 cells ml-1 x to to  $0.98 \times 10^4$  cells ml-1 y, e-increased significantly with the amount

- 338 of visible impurities seen on the ice surface, as shown in Figure 63 ( $F_{2.54}$ =26.1, p<0.0001). No determination of the
- 339 mineralogic composition of the visible impurities was conducted. -A Pearson's product-moment correlation was

340 undertaken conducted to illustrate the relationship between average algal abundance and average DOC and DON

341 concentrations, as DOC and DON concentrations also increased significantly with the amount of visible impurities

- 343 concentrations were significant ( $t_3$ =3.5, p<0.05, r=0.9 and  $t_3$ =5.4, p<0.01, r=0.95, respectively)... Comparison of
- 344 DOP surface ice concentrations and algal cell counts were not significant....
- 345 Dissolved organic nutrient ratios were assessed to investigate the presence of a limiting nutrient—. Molar DON:DOP
- 346 ratios, ranging from 49.3x to 12016.8y, were elevated for all three surface ice environments compared to the 16:1
- Redfield Ratio, and DOC:DOP ratios for all three surface ice habitats, which ranged from 800797.8x to 200013.3y,
- 349 ranged from 1<del>5.6x</del> to 17<del>.2y</del>, were, -only on average, twice2 times</del> the balanced 6.6:1 ratio (Table 1).-.. DON:DOP
- and DOC:DOP ratios also increased with the amount of visible impurities present, while DOC:DON ratios remain
- 351 relatively constant for the three surface ice habitats (Table 1)...
- 352 Fifty four DON samples and 41 DIN samples out of a total of 81 samples for all five supraglacial habitats had
- 353 concentrations above the respective LoD's. Samples resulting below the LoD were considered 0 µM. The field
- 354 blank corrected mean (± standard error) DIN and DON mean concentrations for all five supraglacial environments
- are displayed in Figure 44. Nearly all the DIN was comprised of NH4+, with little to no presence of NO2- or NO3-.
- 356 Overall, mean DON concentrations for the surface ice habitats, which range from 0-14.0 μM, are significantly
- higher (F1.71=12.4, p<0.0001) than mean DIN concentrations, which range from 0–1.1 μM (Fig.ure 44). About 93%
- 358 of the total dissolved nitrogen in all three surface ice habitats was present in the form of DON. Additionally, DON
- 359 concentrations increased significantly from low to medium and low to high visible impurity loadings (F4,71=19.8,
- 360 p<0.05, F4.71=19.8, p<0.001, respectively). T tests revealed significant differences between DON and DIN in all
- 361 supraglacial environments except cryoconite hole water (low:  $t_{36}=3.6$ , p<0.001, medium:  $t_{36}=5.3$ , p<0.0001, high:
- 362 t36=7.4, p<0.0001, stream: t36=-2.6, p<0.01). DON concentrations in cryoconite hole and supraglacial stream water
- 363 fell below the LoD. DON:DOP ratios are elevated for all three surface ice environments compared to the 16:1
- 364 Redfield Ratio (Table 1). DON:DOP ratios also increased with the amount of visible impurities present.
- 365 3.3 Low transport of dissolved organic nutrients within the water table Phosphorus
- 366

367 Mean Dissolved organic nutrient DON and DOP concentrations were decrease significantly lower in supraglacial

368 streams (ranging from a to b0  $\mu$ M and 0.07  $\mu$ M, respectively) and cryoconite hole water (ranging from a to b0.7  $\mu$ M)

- 370 supraglacial stream and cryoconite hole water were significantly lower than ice with high visible impurities
- 371  $(\underline{F}_{4,74} = 6.8, p < 0.001, in both cases)$  (Fig. 6) and a All DON concentrations for cryoconite hole and supraglacial stream
- 372 water were resulted below the LoD (Fig. 34). DIN concentrations were relatively constant over all supraglacial
- habitats with mean concentrations ranging from 0.62 μM to 1.0 μM-. Mean DOP concentrations in supraglacial
- 374 stream  $(0.07 \pm 0.03 \mu M)$  and cryoconite hole water  $(0.07 \pm 0.02 \mu M \mu M)$  were not significantly different from mean

| 375 | DIP concentrations -( $0.07 \mu$ M, $0.01 \mu$ M and $0.01 \pm 0.017 \mu$ M and $0.06 \pm 0.02 \mu$ M, respectively) DIP                         |  |  |  |  |  |  |
|-----|--------------------------------------------------------------------------------------------------------------------------------------------------|--|--|--|--|--|--|
| 376 | concentrations in low ( $0.03 \pm 0.02 \ \mu$ M), medium ( $0.07 \pm 0.02 \ \mu$ M) and high ( $0.05 \pm 0.01 \ \mu$ M) visible impurity ice     |  |  |  |  |  |  |
| 377 | were only slightly elevated compared to supraglacial streams, whereas cryoconite hole water concentrations were                                  |  |  |  |  |  |  |
| 378 | comparable to the three surface ice habitats Mean DOC concentrations in supraglacial stream and cryoconite hole                                  |  |  |  |  |  |  |
| 379 | water (30 µM and 50 µM, respectively which ranged from a to b and c to d respectively) were significantly lower                                  |  |  |  |  |  |  |
| 380 | than ice with high visible impurities (F 4,74 =6.8, p<0.001, in both cases) (Fig. 56). Seventy four DOP samples and 40                |  |  |  |  |  |  |
| 381 | DIP samples out of a total of 81 samples for all five supraglacial habitats had concentrations above the LoD.                                    |  |  |  |  |  |  |
| 382 | Samples resulting below the LoD were considered $0  \mu$ M. The field blank corrected mean (± standard error)                                    |  |  |  |  |  |  |
| 383 | concentrations for all five supraglacial environments are shown in Figure 55. Half of the DIP values fell below the                              |  |  |  |  |  |  |
| 384 | LoD. Mean concentrations for the remaining 40 DIP concentrations ranged from 0-0.07 µM. DOP concentrations                                       |  |  |  |  |  |  |
| 385 | were at least two times higher than the DIP values, with mean DOP values ranging from 0-0.15 $\mu$ M. DOP                                        |  |  |  |  |  |  |
| 386 | concentrations in cryoconite hole and supraglacial stream water fell below the LoD. DOP concentrations were                                      |  |  |  |  |  |  |
| 387 | significantly higher than DIP concentrations in all three surface ice habitats (low: t 36 =3.1, p<0.01, medium: t 36 =2.1, |  |  |  |  |  |  |
| 388 | p<0.05, high: t 36 =3.7, p<0.001)with about 67% of the total dissolved phosphorus present in the form of DOP in all                   |  |  |  |  |  |  |
| 389 | three surface ice habitats.                                                                                                                      |  |  |  |  |  |  |
| 390 | 34 DOC                                                                                                                                           |  |  |  |  |  |  |
|     |                                                                                                                                                  |  |  |  |  |  |  |
| 391 |                                                                                                                                                  |  |  |  |  |  |  |
| 392 | Fifty nine samples out of a total of 81 samples for all five supraglacial habitats had concentrations above the LoD.                             |  |  |  |  |  |  |
| 393 | Samples resulting below the LoD were considered 0 µMDOC concentrations increased with the amount of visible                                      |  |  |  |  |  |  |
| 394 | impurities present in surface ice habitats, as shown in Figure 66, with a significant difference between ice with high                           |  |  |  |  |  |  |
| 395 | and low visible impurity loading ( $F_{4,74}$ =6.8, p<0.01). The field blank corrected mean (± standard error) values for                        |  |  |  |  |  |  |
| 396 | DOC were $83.0 \pm 23.5 \mu$ M, $173 \pm 29.9 \mu$ M and $242 \pm 43.6 \mu$ ML -1 for ice with low, medium and high visible           |  |  |  |  |  |  |
| 397 | impurities, respectively. The corresponding values for supraglacial stream water and cryoconite hole water were                                  |  |  |  |  |  |  |
| 398 | $30.3 \pm 13.5 \mu$ M and $49.6 \pm 33.3 \mu$ M, respectively. DOC concentrations in supraglacial stream and cryoconite hole                     |  |  |  |  |  |  |
| 399 | water were significantly lower than ice with high visible impurities ( $F_{4,74}$ =6.8, p<0.001, in both cases). DOC:DOP                  |  |  |  |  |  |  |
| 400 | ratios for all three surface ice habitats were considerably higher, as much as ~19 times the Redfield ratio, 106:1                               |  |  |  |  |  |  |
| 401 | (Table 1). Yet, DOC:DON ratios were only on average 2 times the balance 6.6:1 ratio (Table 1). DOC:DOP ratios                                    |  |  |  |  |  |  |
| 402 | also increase with the amount of visible impurities present, while DOC:DON ratios remain relatively constant for                                 |  |  |  |  |  |  |
| 403 | the three surface ice habitats (Table 1).                                                                                                        |  |  |  |  |  |  |
|     |                                                                                                                                                  |  |  |  |  |  |  |
| 404 |                                                                                                                                                  |  |  |  |  |  |  |

**405 4. Discussion**

406 4.1 Dominance of dissolved organic phase over dissolved inorganic phase in ice surface environments.

| 407 | Dissolved organic nutrients dominate dissolved inorganic nutrients in the surface ice environments of this region of              |
|-----|-----------------------------------------------------------------------------------------------------------------------------------|
| 408 | the Dark Zone (Fig. 34 ∧ 45). Ninety three percent of the total dissolved nitrogen and ~ 67% of the total                  |
| 409 | dissolved phosphorus found in surface ice habitats was in the dissolved organic phase. To date, this organic phase                |
| 410 | dominance has not been documented in studies of fresh snow or ice cores from the GrIS. As previously mentioned,                   |
| 411 | Telling et al., (2012) reports DIN concentrations in snow found in the margin of the GrIS to be 1.4±0.2 µM L -1 , with |
| 412 | DON concentrations as non-detectableT Furthermore, the comprehensive review conducted by Wolff (2013)                             |
| 413 | states that mean DIN concentrations in ice cores from Greenland are 1.4 µM L 4 , while DON concentrations are also     |
| 414 | non detectible. Furthermore, Wadham et al., (2016) reports elevated DON concentrations in debris rich ice in the                  |
| 415 | Dark Zone of the GrIS during the main ablation season when compared to pre-melt ice and snow. This suggests                       |
| 416 | that potential inputs of nutrients to supraglacial environments, such as fresh snow and melting meteoric ice, are                 |
| 417 | strongly dominated by the dissolved inorganic phase. By contrast, the phase association of dissolved nitrogen at the              |
| 418 | ice surface shifts primarily to the dissolved organic phase during the peak ablation season (July and August). The                |
| 419 | timing of this shift in nitrogen coincides with the reported appearance of the annual Dark Zone and ice algal blooms       |
| 420 | reported byin Tedstone et al., 2017 The timing of the ice algal bloom is is further supported by Williamson et al.,               |
| 421 | (2018) who conducted a transect across the south-west GrIS Dark Zone and documented the extensive and wide-                       |
| 422 | spread algal bloom comprised of pigmented autotrophs during late July and August of 2016 Figure 3_also also                       |
| 423 | elearly shows that algal abundance increases in the ice with low, medium and high visible impurities, suggesting that             |
| 424 | algal cells comprise much of the visible impurities. In fact, Yallop et al., (2012) reported a 3:2 particle to cell ratio         |
| 425 | for surface ice collected in the Dark Zone We therefore hypothesise that the algae present in these blooms drive                  |
| 426 | the shift in nutrients during the peak ablation season from the dissolved inorganic phase to the dissolved organic                |
| 427 | phase.                                                                                                                            |
| 428 | 4.2 Association of dissolved organic nutrients and algal abundance                                                                |
| 429 | CEfficient conversion of dissolved inorganic to Production of dissolved organic nutrients by ice algal assemblages         |
| 430 | was initially is supported by the strong corroboration correlation between average average DON and DOC surface ice                |
| 431 | concentrations and ice algal abundances measured from the same samples <mark>. A closer inspection of the full data se</mark> t   |
| 432 | revealed the presence of a high degree of variability, which caused insignificant relationships between the algal                 |
| 433 | abundance and dissolved organic nutrient concentrations While t he lack of relationship between algal abundance                   |
| 434 | and dissolved organic nutrient concentrations was an unexpected result, the variability was not surprising-                       |
| 435 | Supraglacial environments are dominated by a shallow, 1-2 m, low density porous ice known as the "weathering                      |
| 436 | erust" (Müller and Keeler, 1969;Irvine-Fynn et al., 2012). Due to the intense short wave radiation, the surface of                |
| 437 | supraglacial ice decreases in density and melts internally along grain boundaries, resulting in heterogeneous                     |
| 438 | thickness and porosity (Müller and Keeler, 1969;Cook et al., 2016c;Christner et al., 2018). Supraglacial weathering               |
| 439 | crust has been shown to be extremely dynamic, comprised of infinite flow paths that create an intricate hydrological              |
| 440 | system, interconnecting different habitats and transporting microbes, particles and nutrients (Christner et al., 2018).           |
| 441 | Yet, the flow paths are not always a perfect system for the flow of water due to the differential radiance absorption             |
| 442 | within the ice crystals. In fact, Irvine Fynn et al., (2012) showed that the weathering crust can act as an inhibitor of          |

443 444 nierobes and particles, during times of high discharge. Furthermore, Christner et al., (2018) reported that their 445 predicted values for temperature and water transport in the weathering crust significantly differed with the measured 446 values simply due to the vast number of heterogeneities they did not consider. It is therefore not surprising that over 447 a timescale of two months during the main ablation season in the Dark Zone that the transport of solutes, gases 448 449 450 For example, despite the weak linear association apparent in Figure 7, DOC compared to algal cell counts were 451 significant at the 95% level. The variability within these data is likely driven by the highly dynamic nature of the 452 supraglacial environment. For example, the upper ice surface can be characterised as a perched aquifer, with water 453 percolating through the highly permeable surface ice transporting solutes, gases, organic matter and microbial cells 454 both vertically and horizontally (Irvine-Fynn et al., 2012;Cook et al., 2016c;Christner et al., 2018)W. 455 We still hypothesise interpret these data to demonstrate that ice algal assemblages are the main producers of the 456 dissolved organic nutrient stocks within the melting surface ice of the GrIS, consistent with previous studies of 457 photosynthetic organisms in glacial, freshwater and marine aquatic environments (Johannes and Webb, 458 1970:Lampert, 1978:Musilova et al., 2017). Ice algae that bloom in these environments rapidly uptake inorganic 459 nutrients, which are derived from a number of possible sources, including the atmosphere, wet and dry deposition, 460 and snow and ice-melt (Fig. 7) (Kuhn, 2001: Maccario et al., 2015). This results in an increase in the mass of 461 nutrients held in the microbial biomass, and an increase in dissolved organic nutrients as a by product of the vital 462 intracellular processes and decomposition of the ice algae. An efficient microbial loop, which balances dissolved 463 inorganic nutrient uptake by autotrophic organisms and remineralization by heterotrophic organisms, is often 464 reached in more temperate freshwater aquatic environments (Dodds, 1993). By contrast, work on surface ice near 465 the margin of the GrIS demonstrated bacterial production that was 30 times less than the net primary production of 466 ice algal communities (Yallop et al., 2012). A similar 30:1 ratio was also found by a study conducted in the same 467 study area of the Dark Zone during the 2016 ablation season\_(Nicholes et al., 2019) (Nicholes et al., in review). 468 Dominance of dissolved organic nutrients in surface ice environments highlighted in the present study, in 469 combination with reduced secondary production relative to net primary production in the same environments, 470 indicates an inefficiency inreduced capacity of the microbial loop for remineralization of organic nutrient stocks 471 (Fig. 7) (Yallop et al., 2012; Nicholes et al., 2019)5) (Nicholes et al., in review; Yallop et al., 2012). This assertion 472 is consistent with the findings of previous studies in polar glacier aquatic environments (Stibal et al., 2008a;Stibal et 473 al., 2008b;Stibal et al., 2009;Wadham et al., 2016). For example, as previously stated Stibal et al., (2008) reported 474 that ~70% of the total dissolved nitrogen and ~60% of the total dissolved phosphorus found in supraglacial channel, 475 ervoconite hole and glacier runoff environments of a Svalbard glacier were in the dissolved organic phase. Wadham 476 et al., (2016) found elevated DON concentrations in cryoconite holes and debris rich ice relative to snow and pre-477 melt ice in the Dark Zone of the GrIS. They hypothesised that the elevated DON concentrations were caused by 478 either mineralization of organic matter by microbial activity or leaching of allochthonous organic matter in debris.

479 Furthermore, Stibal et al., (2008) reported that ~72% of the TDN and ~89% of the TDP found in cryoconite holes on

- 480 a Svalbard glacier were in the dissolved organic phase. This suggests that conversion of dissolved inorganic to
- 481 dissolved organic nutrients by autotrophs in melting surface ice environments may be a common process on many

482 <mark>glacier surfaces</mark>.

483 4.3 Retention of nutrients at ice sheet surface

484 The intense solar radiation received by glacier and ice sheet surfaces produces internal melting and density reduction 485 within the near surface ice, resulting in a unique porous surface ice layer also known as the weathering crust 486 (LaChapelle, 1959; Müller and Keeler, 1969; Munro, 1990). The porous nature of the weathering crust allows flow 487 paths to form through the water table that exists within the surface ice (Irvine Fynn et al., 2012;Cook et al., 488 2016c;Rassner et al., 2016;Christner et al., 2018). These flow paths serve as important links between different 489 supraglacial environments and are believed to transport microbes and nutrients via subsurface flow (Irvine Fynn et 490 al., 2012;Hoffman et al., 2014;Karlstrom et al., 2014;Cook et al., 2016e). Overall, the DON and DOC in 491 supraglacial streams and cryoconite hole water were lower than the DON and DOC in all surface ice habitats and 492 significantly lower than the surface ice with high visible impurities (Fig.ures 44 & and 66). Our data, therefore, 493 likely indicate a retention of organic nutrient phases within surface ice environments. One mechanism of possible 494 retention is the production of extracellular polymeric substances (EPS). Algae and bacteria produce EPS which can 495 alter the physical and chemical environment around their cells (Stibal et al., 2012a; Angelaalinev et al., 2017). For 496 example, it has been shown that EPS are used by eyanobacteria in ervoconite holes to bind mineral particles together 497 ereating the cryoconite granules at the bottom of the hole (Stibal et al., 2012b; Yallop et al., 2012; Musilova et al., 498 2016). EPS exists in the colliodal form and when analysed from melted surface ice samples, it is likely constrained 499 in the dissolved organic fraction (Pereira et al., 2009;Hodson et al., 2010). Yet, it is possible that this retention is 500 transitory, and ice surface habitats have the potential to supply a large pulse of dissolved organic nutrients to 501 downstream ecosystems. For example, Musilova et al., (2017) reported that at the margin of the GrIS, DOC 502 remaining in surface ice at the end of the ablation season likely froze over winter and was released the following ablation season through ice melt. Furthermore, Wadham et al., (2016) produced a time series of DON 503 504 concentrations in runoff from Leverette glacier, a terminating glacier on the GrIS, showing the highest concentration 505 in early May and decreasing throughout the main melt season. The enrichment of DON concentrations also reported 506 by Wadham et al., (2016) for moulin water in the Dark Zone, suggests acquisition of DON from supraglacial 507 environments while the elevated DON concentrations in runoff water from the base of Leverette Glacier, compared 508 to snow and pre-melt iee during the main melt season, suggest transport of this supraglacial DON to downstream 509 environments . (Wadham et al., 2016)This supports the hypothesis of dissolved organic nutrients being retained at 510 the ice surface over winter and coincidently supplying a large pulse of dissolved organic nutrients at the onset of the 511 following melt season. Yet, For example, Musilova et al., 2017 reported that at the margin of the GrIS, DOC 512 remaining in surface ice at the end of the ablation season likely froze over winter and was released the following 513 <mark>a</mark>blation season through ice melt. tTheproportional input of dissolved organic nutrients in downstream export of 514 DOM from supraglacial environments in the Dark Zone of the GrIS is currently unknown.

|     | v i o                                                                                                                   |
|-----|-------------------------------------------------------------------------------------------------------------------------|
| 516 | Carbon, nitrogen and phosphorus are required by all cells for balanced growth. The generalised stoichiometry for        |
| 517 | C:N:P in marine phytoplankton, the Redfield Ratio, is 106:16:1 (Redfield, 1958). It is important to note, however,      |
| 518 | that while the Redfield Ratio is commonly used as the main stoichiometry reference, it is a specific ratio for marine   |
| 519 | aquatic environments only. Differing stoichiometries have been reported for diverse environments. For example,          |
| 520 | Barrett et al., (2017) investigated different environments in the Dry Valleys of Antarctica and found average N.P       |
| 521 | ratios for surface ice and snow-environments and eryoconite holes on glaciers to be 21:1 and 15:1, respectively-        |
| 522 | The average N:P ratios in the same Dry Valley site for streams and takes fed by glacier melt were 12:1 and 25:1,        |
| 523 | respectively. The variability and changes in N:P ratios over time were caused mainly by the presence and activity       |
| 524 | of microorganisms in the environment and the geochemical availability of nitrogon and phosphorus in the area s          |
| 525 | Furthermore, Latz et al., 2017 investigated the particulate C:N:P ratios of snow and ice habitats in Sweden and         |
| 526 | Svalbard. They found high particulate C:N and low-particulate N:P ratios, which they concluded as likely N-             |
| 527 | limitation rather than a more common P-limitation.                                                                      |
| 528 | Have we avarage the DOC DON DOP ratios of maked surface is complex in an atterant to determine the limiting             |
| 520 | nutriant of supraglasial anticonments in the Dark Zone. The dissolved organic CNUP estics reported for our surface      |
| 520 | the complex are notably kicker than the Radfield Ratio, indicating that the system could be R limited. For example,     |
| 530 | DON-DOR (49-78-116) and DOC-DOR (707-1166, 2013) ratios reported respectively for low-medium and high                   |
| 532 | surface ice anxientments are extremely birb compared to their 1611 and 10611 Redfield ratio counterness (Table 1)       |
| 532 | They also increase as the amount of visible impurities increase. In conteast, DOC-DON ratios are on average only        |
| 534 | two times higher than the Redfield ratio of 6.6.1 (Table 1) DOC DOP and DON DOP ratios increase with the                |
| 535 | amount of visible impurities, as at a creater rate than DOC DON ratios remain relatively stable for surface ice         |
| 536 | habitaty. This indicates that the more algal biomass present, the higher the retention of DOP, in order to achieve and  |
| 537 | maintain homeostasis, compared to DON and DOC (Table 1), suggesting that P limitation increases with higher             |
| 538 | algal biomass loading in surface ice habitats.                                                                          |
|     |                                                                                                                         |
| 539 | High DOC:DOP and DON:DOP ratios have been documented in other glacial polar aquatic environments. Stibal ei             |
| 540 | al., (2008) showed that DOC:DOP ratios were10 times higher than the Redfield ratio on a Svalbard glacier and            |
| 541 | that DON:DOP ratios exceed the balanced ratio by a factor of three. This is not entirely surprising as P is a rock-     |
| 542 | dorived minoral that is only released into the dissolved phase by chemical and physical weathering. When compared       |
| 543 | to alpine glaciers, ice sheet surface environments receive less lithological debris via terrestrial and atmospheric     |
| 544 | processes, due to their relative proximity to source material. It is, therefore, reasonable for dissolved phosphorus to |
| 545 | be the limiting nutrient compared to nitrogen and carbon, both of which are more readily available from the             |
| 546 | atmosphere:                                                                                                             |
| 547 | Cryceonite, a rock derived substance with a high organic carbon content, is found in abundance on many polar ice        |
| 548 | authors and cover 0.5% of the author iss in the oblation zone of the Cals (Cribbon, 1070, Stibul et al.                 |
| 5-0 | and the second station of the second second and an analysis for the second (second second second                        |
| 549 | 2012b;Bagshaw et al., 2013;Cook et al., 2016a;Ryan et al., 2018). Stibal et al., (2008) investigated the potential      |
| 1   |                                                                                                                         |

**515 4.4 Stoichiometry of different supraglacial environments**

| 550 | bioavailability of phosphorus from cryoconite in cryoconite holes on a Svalbard glacier and found the potentially                |
|-----|----------------------------------------------------------------------------------------------------------------------------------|
| 551 | bioavailable pool of phosphorus in oryoconite to be -0.16mg g -1 - Furthermore, Lutz et al., 2017 investigated the    |
| 552 | particulate C.N:P ratios of snow and ice habitats in Sweden and Svalbard. They found high particulate C:N and low                |
| 553 | particulate N.P ratios, which they concluded as likely N limitation rather than a more common P limitation. This                 |
| 551 | answer that the minimized and any added to assess the marken late R mark multiplication the martinulate N. While                 |
| 554 | suggest the one increditions? Was able to decess the particulate r more reacht than the particulate (s) white                    |
| 555 | investigations into the targeted ability of microbes to utilize this particulate inorganic phosphorus pool have yet to be |
| 556 | conducted, Tedstone et al., (2017) noted that widespread ice algal blooms may only occur where abundant                          |
| 557 | particulates are available as they could be providing necessary nutrients for the ice algal assemblages. Clearly,                |
| 558 | further investigation into the influence of particulate phospherus sources and utilization is needed to fully                    |
| 559 | understand the nutrient cycle accurring in supraglacial environments as the dissolved nutrient input might only                  |
| 560 | the second se                  |
| 500 | represent of portion of the oxiganing system. The minimum end of thissoffeet or game over dissoffeet morganic 
[revised manuscript text omitted]
 ice algal community, since the DOP produced by heterotrophic activity and/or as EPS has decreaseds.
- 673 However, this does not quite tie in with the DIP data presented in Fig. 4-4, which shows that measurable, if low,
- 674 concentrations of P are usually present in the melting surface ice. Rather, NO3 and NO2 are below detection,
- 675 presumably as a result of uptake by phototrophs, and  $NH_4^+$  is the only measurable DIN species, presumably as a
- 676 result of heterotrophic activity. Phototrophs-are usually thought to favour preferentially the uptake of utilize both
- 677  $\underline{NH_4^+}$  and  $\underline{over}$  NO3, and the presence of both DINP and DIP N-in the melting surface ice environments, irrespective
- 678 of visible particulate loading, and therefore of algal cell <del>counts</del>abundance, suggests that a factor other than
- 679 macronutrient concentration is limiting algal growth. Table 1 shows that mean  $NH_4^+$  concentrations in the melting
- 680 surface ice are in the range of  $0.6 1.0 \mu M \pm^4$ . We noted above that there is no readily available C:N ratio of glacier
- 681 ice-algae in the literature, but typical C:N ratios of sea ice algae are in the range of 12-46 (Niemi and Michel, 2015).
- **682** This implies that somewhere in the range of  $7.2 26 \mu M \pm^{-1}$ -of C could be additionally fixed, if all the N wasere to
- taken up by phototrophs with the this range of C:N ratios. We also noted that it is even more difficult to find C:N:P
- ratios of glacier iee-algae, but should the C:P ratio be in the region of 100:1 to 1000:1, then P demand will be 0.007
- $\frac{-0.46 \,\mu\text{M}}{\pm}$  Table 1 shows that the mean concentration of DIP in melting surface ice is in the range of 0.03 to
- 686 0.05 μM, which suggests that P is not a limiting macronutrient on primary production. The systematic change in the

687 DON:DOP and DOC:DOP ratios with increasinge in-visible impurities, which is a proxy for algal cell

688 countsabundance, could be driven by the amount of P per cell that is potentially available at the high light intensity

 $\frac{\text{of the ablation season} (> 1500 \,\mu\text{mol photons m}^2 \,\text{s}^{-1}). \text{ The DIP content of the surface ice is relatively constant (Table}{}$

690 1) given the much larger change in cell <del>counts</del>abundance as the visible impurities increase. The combination of

691 lower P availability at high light intensity results in an increase in the C:P ratio of phototrophs in other aquatic

692 environments (Hessen et al., 2013). (xx). It is plausible that this too happens with glacier ice algae, and that

693 subsequent decomposition products and EPS will likewise have higher DOC:DOP ratios as a consequence...

694

695

**696 5. Conclusion**

697 We conclude that DIN and DON -concentrations in the melting surface ice of the Dark Zone on the GrIS are 698 markedly different from those documented in ice cores to date. Wolff et al., 2013 reported DIN, principally in the 699 form of NO3, dominatinges the initial composition of ice melt{Wolff, 2013 #32}, yet in the present study, DON 700 dominates tin-the melting surface ice evironments environments which host blooming glacier ice-algae. 701 TheFurthermore, ice algal assemblDIN in these environments is exclusively present ast  $NH_4^+$ , and  $NO_3^-$  is below the 702 detection limit (0.64 µM)xx).-. ages that bloom in the Dark Zone of the GrIS during the ablation season are the 703 main drivers of the nutrient cycling occurring in melting surface ice environments. There is relatively little data on 704 the P content of Greenland ice, but we find that DOP dominates DIP in the melting surface ice habitats, although 705 DIP is usually present in measurable quantities (the detection limit is  $LoD = -0.02 \ \mu M_{\text{MM}}$ ).... The presence of both 706 NH4+ -and DIP, even in even-heavily colonised melting surface ice, suggests that factors other than macronutrient 707 limitation control the blooms. We speculate that dissolved macronutrients are held in the melting surface ice because 708 of the architecture of the weathering crust, and/or because EPS is retained within the melting ice latticework. The 709 former controls the hydrology and the connectivity of water flow paths and water storage in the surface ice, and the 710 latter may be involved with the retention of particulates in the surface. There is currently no data on the C:N:P ratios 711 of the EPS exuded by glacier ice algae, but the EPS of other autotrophs does contain both N and P in association 712 with C. -The DOC:DON ratios areis relatively constant in the melting surface ice, but the DOC:DOP ratios increases 713 markedly with increasing algal cell counts. This may be attributable to the increasingly higher cells to DIP ratio, 714 which, at high light intensity, increases the C:P ratio of autotrophs in other freshwater environments (Hessen et al., 715 2013).—. This could be seen as ais a beneficial adaption to algal life in melting ice surfaces, where P sources are 716 limited, since blooms are not so dependent on P as a consequence, should this adaptation also be found in glacier ice 717 algae. Our data indicates a rapid uptake of available dissolved inorganic nutrients and a high production of dissolved 718 organic carbon, nitrogen and phosphorus. The relatively high concentrations of dissolved organic nutrients found on 719 the ice surface, combined with reduced secondary production relative to net primary production, suggests an 720 inefficient or inhibited microbial loop for the remineralization of organic nutrient stocks (Yallop et al., 721 2012;Nicholes et al., 2019) (Nicholes et al., accepted). Furthermore, the contrast in dissolved organic nutrient 722 concentrations in surface ice environments compared to supraglacial streams and cryoconite hole water point to

| 724
725 | nutrient pool, and its adhesive properties. This retention could result in supraglacial environments acting as large
sources of dissolved organic nutrients for downstream ecosystems during the onset of the following ablation
season., Yyet, the proportion of DOM export from supraglacial environments of the Dark Zone compared to DOM |
|------------|-----------------------------------------------------------------------------------------------------------------------------------------------------------------------------------------------------------------------------------------------------------------------------------------------------------------------------------------------------------|
| 725        | sources of dissolved organic nutrients for downstream ecosystems during the onset of the following ablation
season. , Yyet, the proportion of DOM export from supraglacial environments of the Dark Zone compared to DOM                                                                                                                        |
|            | season., Yyet, the proportion of DOM export from supraglacial environments of the Dark Zone compared to DOM                                                                                                                                                                                                                                               |
| 726        |                                                                                                                                                                                                                                                                                                                                                           |
| 727        | inputs from subglacial processes in outlet glaciers requires further research. export of DOM from the Dark Zone it                                                                                                                                                                                                                                        |
| 728        | <del>is still unknown.</del>                                                                                                                                                                                                                                                                                                                              |
|
729    |                                                                                                                                                                                                                                                                                                                                                           |
| 730        |                                                                                                                                                                                                                                                                                                                                                           |
|
731    | Data Availability                                                                                                                                                                                                                                                                                                                                         |
| 732        | All data will be made available upon acceptance and publication of the articleData will be inputted into an open                                                                                                                                                                                                                                          |
| 733        | access file.                                                                                                                                                                                                                                                                                                                                              |
| 734        |                                                                                                                                                                                                                                                                                                                                                           |
| 735        | Acknowledgments                                                                                                                                                                                                                                                                                                                                           |
| 736        | The authors would like to thank and acknowledge the entire Black & Bloom team, especially those involved in the                                                                                                                                                                                                                                           |
| 737        | sample collection conducted in the 2016 field season The manuscript was considerably improved following the                                                                                                                                                                                                                                               |
| 738        | constructive commentary of two anonymous reviewers.                                                                                                                                                                                                                                                                                                       |
| 739        |                                                                                                                                                                                                                                                                                                                                                           |
| 740        | Team List                                                                                                                                                                                                                                                                                                                                                 |
| 741        | Liane G. Benning (GFZ German Research Centre for Geosciences, Potsdam, Germany), James B. McQuaid                                                                                                                                                                                                                                                         |
| 742        | (University of Leeds, Leeds, UK), Andrew J. Hodson (University Centre in Svalbard, Longyearbyen, Norway &                                                                                                                                                                                                                                                 |
| 743        | Western Norway University of Applied Sciences, Bergen, Norway), Edward Hanna (University of Lincoln, Lincoln,                                                                                                                                                                                                                                             |
| 744        | UK), Tristram D. L. Irvine-Fynn (Aberystwyth University, Aberystwyth, UK), Jonathan L. Bamber (University of                                                                                                                                                                                                                                              |
| 745        | Bristol, Bristol, UK), Stefanie Lutz (GFZ German Research Centre for Geosciences, Potsdam, Germany), Miranda J                                                                                                                                                                                                                                            |
| 746        | Nicholes (University of Bristol, Bristol, UK), Marek Stibal (Charles University, Prague, Czech Republic), Jason E.                                                                                                                                                                                                                                        |
| 747        | Box (Geological Survey of Denmark and Greenland, Copenhagen, Denmark).                                                                                                                                                                                                                                                                                    |
| 748        |                                                                                                                                                                                                                                                                                                                                                           |
| 749        | Author contribution                                                                                                                                                                                                                                                                                                                                       |
| 750        | MT, AA and MY conceived and designed the studyAH, CW, MT, AA, AT, JM, JC and the Black & Bloom                                                                                                                                                                                                                                                            |
| 751        | group collected the samplesCW provided algal counts for the mid to late ablation periodsAH conducted all the                                                                                                                                                                                                                                              |

| 752
753 | nutrient analysis and was aided by FS in the instrument maintenance and data analysis.—.AH wrote the paper with inputs from MT, CW, AT and AA.—.All authors reviewed the final manuscript.—. |
|------------|----------------------------------------------------------------------------------------------------------------------------------------------------------------------------------------------|
| 754        |                                                                                                                                                                                              |
| 755        | Competing Interests                                                                                                                                                                          |
| 756        | The authors declare they have no conflicts of interest.                                                                                                                                      |
| 757        |                                                                                                                                                                                              |
| l
758   |                                                                                                                                                                                              |
| 759        | Funding                                                                                                                                                                                      |
| 760        | This project has received funding from the European Union's Horizon 2020 research and innovation programme                                                                                   |
| /61        | under the Marie Sklodowska-Curie grant agreement No 675546—. This work was also funded in part by the UK                                                                                     |

under the Marie Sklodowska-Curie grant agreement No 675546—. This work was also funded in
Natural Environment Research Council Consortium Grant 'Black and Bloom' (NE/M0212025).